# Sensorimotor delays constrain robust locomotion in a 3D kinematic model of fly walking

Lili Karashchuk[1,2,3†§], Jing Shuang Li[4†], Grant M Chou[2], Sarah Walling-Bell[2], Steven L Brunton[5], John C Tuthill[2*‡], Bingni W Brunton[3*‡]

[1]Neuroscience Graduate Program, University of Washington, Seattle, United States; [2]Department of Neurobiology and Biophysics, University of Washington, Seattle, United States; [3]Department of Biology, University of Washington, Seattle, United States; [4]Department of Electrical Engineering and Computer Science, University of Michigan, Ann Arbor, United States; [5]Department of Mechanical Engineering, University of Washington, Seattle, United States

*For correspondence:
tuthill@uw.edu (JCT);
bbrunton@uw.edu (BWB)

†These authors contributed equally to this work
‡These authors also contributed equally to this work

Present address: §Allen Institute for Neural Dynamics, Seattle, United States

## eLife Assessment

This **valuable** study presents a computational model that simulates walking motions in *Drosophila* and suggests that, if sensorimotor delays in the neural circuitry were any longer, the system would be easily destabilized by external perturbations. The hierarchical control model is sensible and the evidence supporting the conclusions **convincing**. The modular model, which has many interacting components with varying degrees of biological realism, will serve as a well-grounded starting point for future studies that incorporate richer or more complete empirical data.

**Abstract** Walking animals must maintain stability in the presence of external perturbations, despite significant temporal delays in neural signaling and muscle actuation. Here, we develop a 3D kinematic model with a layered control architecture to investigate how sensorimotor delays constrain the robustness of walking behavior in the fruit fly, *Drosophila*. Motivated by the anatomical architecture of insect locomotor control circuits, our model consists of three component layers: a neural network that generates realistic 3D joint kinematics for each leg, an optimal controller that executes the joint kinematics while accounting for delays, and an inter-leg coordinator. The model generates realistic simulated walking that resembles real fly walking kinematics and sustains walking even when subjected to unexpected perturbations, generalizing beyond its training data. However, we found that the model's robustness to perturbations deteriorates when sensorimotor delay parameters exceed the physiological range. These results suggest that fly sensorimotor control circuits operate close to the temporal limit at which they can detect and respond to external perturbations. More broadly, we show how a modular, layered model architecture can be used to investigate physiological constraints on animal behavior.

## Introduction

Animals as diverse as tardigrades (*Nirody et al., 2021*) and tapirs (*Catavitello et al., 2018*) use inter-limb coordination to walk through complex terrain. When a walking animal encounters an unexpected perturbation (e.g. it is pushed or tripped), its ability to recover and sustain locomotion can be a matter of life and death. However, the presence of significant temporal delays in animal sensorimotor systems establishes fundamental limits on how quickly animals can respond to external perturbations

(*More and Donelan, 2018*; *Gebehart and Büschges, 2021*). Sources of temporal delay include neural conduction, synaptic transmission, and electromechanical muscle activation (*Sterling, 2015*; *Ijspeert and Daley, 2023*).

Sensorimotor delays pose a particular challenge for systems that rely on feedback control, because they limit the availability of up-to-date information on the state of the external environment (*Franklin and Wolpert, 2011*). For example, time delays limit robust performance in feedback control systems (*Doyle et al., 2013*), and model predictive control (MPC) (*García et al., 1989*; *Camacho, 2013*; *Brunton, 2022*) is one algorithmic strategy to improve performance while maintaining robustness. In biological systems, theoretical studies suggest that some animals use predictive internal models to mitigate the effect of delays (*More and Donelan, 2018*; *Li et al., 2023*; *Desmurget and Grafton, 2000*). Limb compliance and other biomechanical adaptations can also compensate for unexpected perturbations within a limited range and mitigate the effect of sensorimotor delays (*Daley, 2018*; *Ashtiani et al., 2021*). However, the constraints that sensorimotor delays impose on robust locomotion have been difficult to quantify, because it is experimentally challenging to manipulate sensorimotor delays and observe their effects on animal locomotion (although this has been achieved in other motor systems, for example using focal cooling in singing birds *Banerjee et al., 2021*). Computational models of locomotion typically have not included delays as a tunable parameter, although some models have included them as fixed values (*Geyer and Herr, 2010*; *Geijtenbeek et al., 2013*). In general, the impact of sensorimotor delays on locomotor control and robustness remains underexplored in computational neuroscience.

Flies are agile and robust walkers, and the availability of genetic and behavioral tools in *Drosophila* makes them well-suited to investigate neural mechanisms of locomotor control (*DeAngelis et al., 2019*; *Gonçalves et al., 2022*; *Cruz and Chiappe, 2023*). The fly is also the only walking animal whose nervous system is almost completely mapped at synaptic resolution (*Galili et al., 2022*). Flies walk rhythmically with a continuum of stepping patterns that range from tetrapod (where two of six legs are off the ground at a time) to tripod (where three of six legs are off the ground at a time) (*DeAngelis et al., 2019*; *Mendes et al., 2013*; *Szczecinski et al., 2018*; *Wosnitza et al., 2013*; *Strauss and Heisenberg, 1990*; *Nishii, 2000*; *Pratt et al., 2024*). Each fly leg has five joints that move through seven mechanical degrees of freedom (*Karashchuk et al., 2021*; *Lobato-Rios et al., 2022*) and are actuated by ~18 muscles that are innervated by ~70 motor neurons (*Azevedo et al., 2024*). Each leg motor neuron is uniquely identifiable (*Azevedo et al., 2024*) and receives thousands of synaptic inputs from hundreds of unique premotor neurons within the fly's ventral nerve cord (VNC, *Lesser et al., 2024*; *Cheong et al., 2024*), a part of the invertebrate nervous system analogous to the vertebrate spinal cord. The architectural features of *Drosophila* locomotor control—joints, leg motor neurons, and the connectivity of the VNC—motivates the architecture of our proposed model.

Past models of insect multi-legged walking have taken three general approaches. The first approach models legs as coupled oscillators or inverted pendula without taking into account the mechanics of leg joints. Each leg is a single oscillator, and a network of oscillators is tuned to recreate an oscillatory gait from measurements of footfalls (*Couzin-Fuchs et al., 2015*; *Proctor and Holmes, 2018*); alternatively, tripod gaits are approximated by spring-loaded inverted pendula (*Chun et al., 2021*). The second approach focuses on the physical details of the legs and their joints. *Schilling et al., 2013* use a decentralized reactive controller to recreate hexapod walking patterns; *Goldsmith, 2019* and *Goldsmith et al., 2020* introduce a robotic platform; *Lobato-Rios et al., 2022*, *Wang-Chen et al., 2023*, and *Vaxenburg et al., 2024* develop virtual fly simulations in a physics engine. Walking behavior in these models is also driven by coupled oscillators with tuned parameters that reproduce inter-leg coordination patterns; however, the resultant joint kinematics are typically unrealistic. The third approach uses normative learning and optimization to generate walking behaviors de novo. *Ramdya et al., 2017* maximized fly walking speed by varying inter-leg coordination with a genetic algorithm, whereas (*Heess et al., 2017*) maximized walking speed in both bipedal and quadruped walkers with reinforcement learning. Normative models produce walking with varying degrees of realism, but they require a clever selection of objectives with constraints and are computationally expensive. Furthermore, any change in delay values would require a computationally intensive retraining process. Notably, *Geijtenbeek et al., 2013* include fixed sensorimotor delays in a bipedal model, but do not explore the effect of varying delay values. Overall, existing walking models focus on either kinematic or physiological accuracy, but few achieve both, and none consider the effect of varying sensorimotor delays.

Here, we develop a new, interpretable, and generalizable model of fly walking, which we use to investigate the impact of varying sensorimotor delays in *Drosophila* locomotion. The model features several key innovations. First, the model is trained on high-resolution 3D joint angle data from walking flies, which has only recently become possible due to new deep learning-based computer vision tools (*Karashchuk et al., 2021*). Second, the model's multi-layered architecture achieves more than the sum of its parts: specifically, a neural network model recapitulates kinematic coordination of many joints, and an optimal controller allows the data-driven model to generalize to new scenarios (e.g. large delays and perturbations) without retraining. Third, the inclusion of delay as a tunable parameter allows us to systematically investigate the quantitative relationships between sensorimotor delays and robust walking. Fourth, we introduce a new method to quantitatively compare the kinematic similarity of real and simulated walking. Overall, these analyses suggest that neuromuscular delays limit how fast flies can walk while retaining robustness to unexpected perturbations. They also illustrate a general approach to use in silico experiments with virtual animals to investigate how fundamental physiological parameters constrain animal behavior.

## Results

## A kinematic model of fly walking that incorporates delays and accommodates perturbations

We designed a walking model with three functional layers (*Figure 1B*), inspired by the hierarchical anatomical organization of the fly nervous system (*Figure 1A*, *Dallmann et al., 2021*). The three layers are an inter-leg *phase coordinator*, a *trajectory generator*, and an *optimal controller* that interfaces with a leg dynamics model. Each individual leg is modular and governed by its own dynamics, optimal controller, and trajectory generator. Inter-leg coordination is accomplished by the phase coordinator alone. In other words, the movement of each leg is not coupled to any other leg except through the phase of its current step cycle. This modularity is inspired by the segmental neuroanatomy of the VNC, in which each leg is controlled by distinct local premotor circuits and pools of motor neurons (*Lesser et al., 2024*; *Cheong et al., 2024*). While the model is inspired by neuroanatomy, its components do not strictly correspond to components of the nervous system—the construction of a neuroanatomically accurate model is deferred to future work (see Discussion).

Each layer in the model is an abstraction for the layer below it, such that various elements of walking (e.g. joint control, inter-leg coordination) can be integrated through modules. Below, we describe each modular component of our model; more details on its derivation and implementation are elaborated in the Methods and materials and Appendices. Although the model can turn and side-step, our analysis focuses on forward-walking — the model is driving the fly to walk straight in all simulations, unless otherwise stated.

### Inter-leg phase coordinator

To coordinate multi-legged walking, we modeled the step phase of each leg as an oscillator. We refer to the left and right legs as L1–3 and R1–3, respectively, where the front legs are L1 and R1. The phase coupling of the six-leg oscillators establishes realistic inter-leg coordination. We use a Kuramoto oscillator model (*Acebrón et al., 2005*; *Strogatz, 2000*) to perform this coordination, as in *Proctor and Holmes, 2018*; equations and implementation details are in Methods and materials. Briefly, the phase coordinator takes the instantaneous phases $\phi$ from all legs as input, then outputs the desired phases $\phi_d$ to all legs. These desired phases are synchronized across pairs of legs to maintain a tripod coordination pattern, even when subject to unpredictable perturbations. We estimated the phase coupling coefficient among legs from measured 3D joint kinematics of walking flies (*Karashchuk et al., 2021*). Without the phase coordinator, individual trajectory generators would generate realistic kinematics for each leg, but they would not be coordinated with each other.

### Joint kinematics trajectory generator

The trajectory generator layer is responsible for producing realistic 3D joint kinematics of each leg. To express the relationships among all leg joint angles, we use an artificial neural network trained to generate angle trajectories. The modular design makes it possible to train trajectory generators separately from the inter-leg phase coordinator and optimal controller. Ultimately, the optimal controller

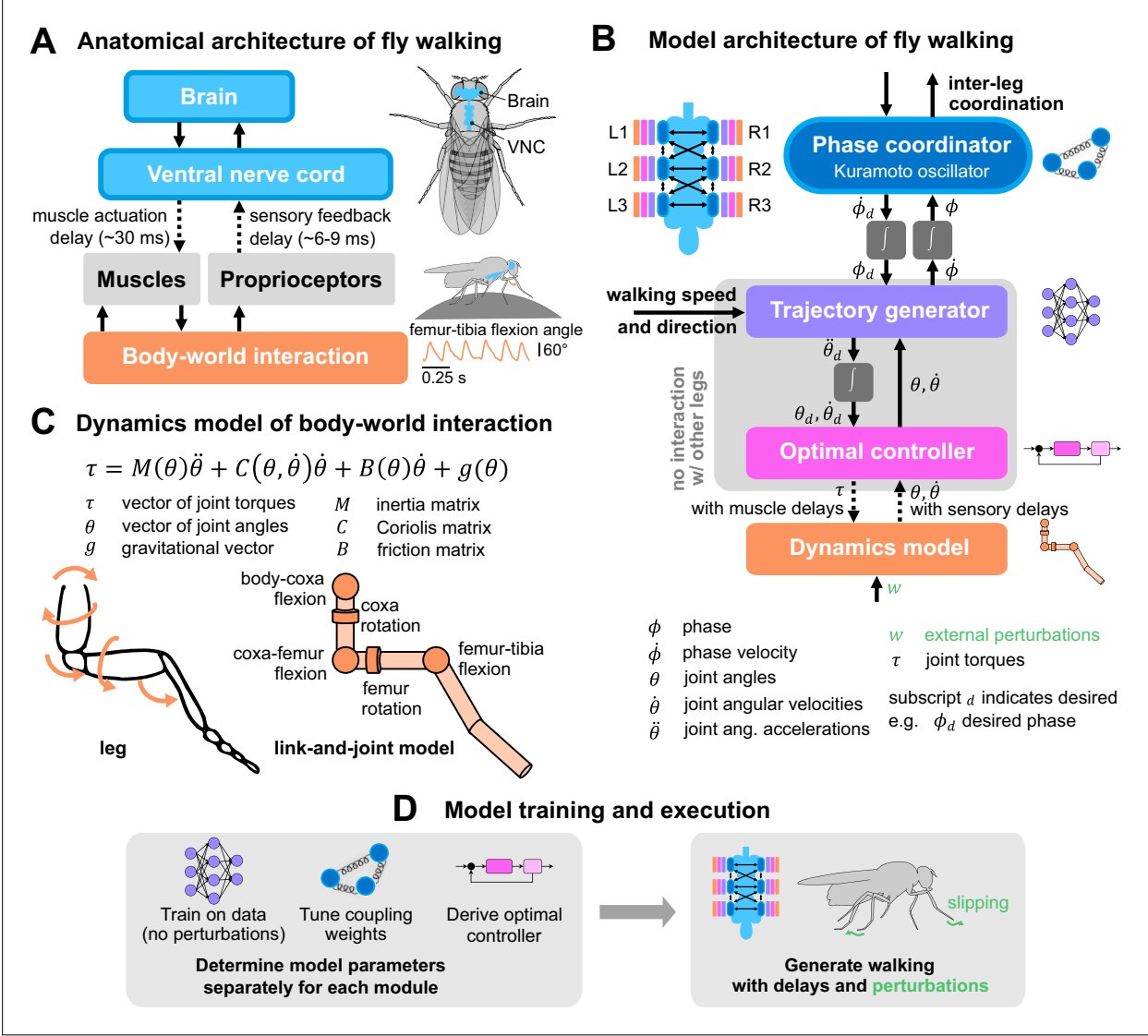

**Figure 1.** A layered model of fly walking that incorporates link-and-joint dynamics and sensorimotor delays. (**A**) The neural systems that control walking are hierarchically organized. The brain sends high-level commands (e.g. walking speed and direction) to the ventral nerve cord (VNC). Approximately 70 motor neurons (MNs, **Azevedo et al., 2024**) control each leg. The cell bodies and dendrites of the MNs are in the VNC, and their axons innervate muscles in the leg. Each leg muscle is actuated, after some delay, following activation by one or more motor neurons. Sensorimotor circuits in the VNC also receive delayed sensory feedback via proprioceptors. (**B**) Multi-layered walking model architecture. Body-world interactions are mediated by a dynamics model. Proprioceptive feedback consists of joint angles and angular velocities (**Mamiya et al., 2018**). Each leg has an optimal controller operating at 600 Hz that interfaces with the dynamics model and a trajectory generator that generates realistic kinematics. The trajectory generator is learned from data and operates at 300 Hz. It interfaces with the phase coordinator, a Kuramoto oscillator that maintains inter-leg coupling. The trajectory generator and optimal controller mimic local circuits within each leg, so they do not interact with other legs. Phase coupling is the only information shared between all legs. (**C**) Dynamics model of a fly leg, derived from link-and-joint models and Euler-Lagrange equations. (**D**) Schematic of model training and execution process. Model parameters for each module are tuned independently, then the modules are assembled to generate walking with delays and in response to unexpected perturbations.

integrates desired angle trajectories with proprioceptive joint angle feedback to output joint torques to the physical model of each leg.

As illustrated schematically in *Figure 1B*, the inputs to the trajectory generator module are desired leg phase $\phi_d$, joint angles $\theta$, joint angular velocities $\dot{\theta}$, and walking speed and direction of the fly. The trajectory generator then outputs the leg phase velocity $\dot{\phi}$ and desired angular accelerations $\ddot{\theta}_d$. This output is integrated to produce the desired angle and angular velocities $\theta_d, \dot{\theta}_d$, which are the inputs to the optimal controller.

To train the network from data, we used joint kinematics of flies walking on a spherical treadmill, obtained from tracking 3D joint angles with Anipose (*Karashchuk et al., 2021*). Details on training approach and network properties are described in Methods and materials. The walking speed and directions are signals that are not generated from other modules of the model, but are instead external inputs to the trajectory generator computed from the data; biologically, these signals are analogous to descending signals from the fly brain (*Simpson, 2024*). This organization is motivated by the observation that walking velocity and direction have a substantial effect on joint angles, but they do not have a substantial effect on the parameters and outputs of the phase coordinator (e.g. phase offsets), as substantiated in Appendix 1.

After training and when assembled with the other layers of the model, the trajectory generator receives proprioceptive information on the observed, current state of the leg $\theta, \dot{\theta}$, as well as the target desired phase $\phi_d$ from the phase coordinator. Thus, it generates a time series of desired angles and angular velocities for some future interval and sends this time series $\ddot{\theta}_d$ to the controller; it also estimates the current phase velocity $\dot{\phi}$ of the leg, which is passed to the phase coordinator.

In the absence of external perturbations, the trajectory generator produces realistic joint angles similar to those of walking flies, as we show below. When challenged with unpredictable external perturbations, the impact on the trajectory generator is mitigated by the optimal controller layer, which attempts to return the actual state to the desired state. This control is possible because the controller operates at a higher temporal frequency than the trajectory generator in the model. The controller can perform many iterations (and reject disturbances) in between updates to and from the trajectory generator. We emphasize that all data used to train the trajectory generator came from experimental conditions without external perturbations.

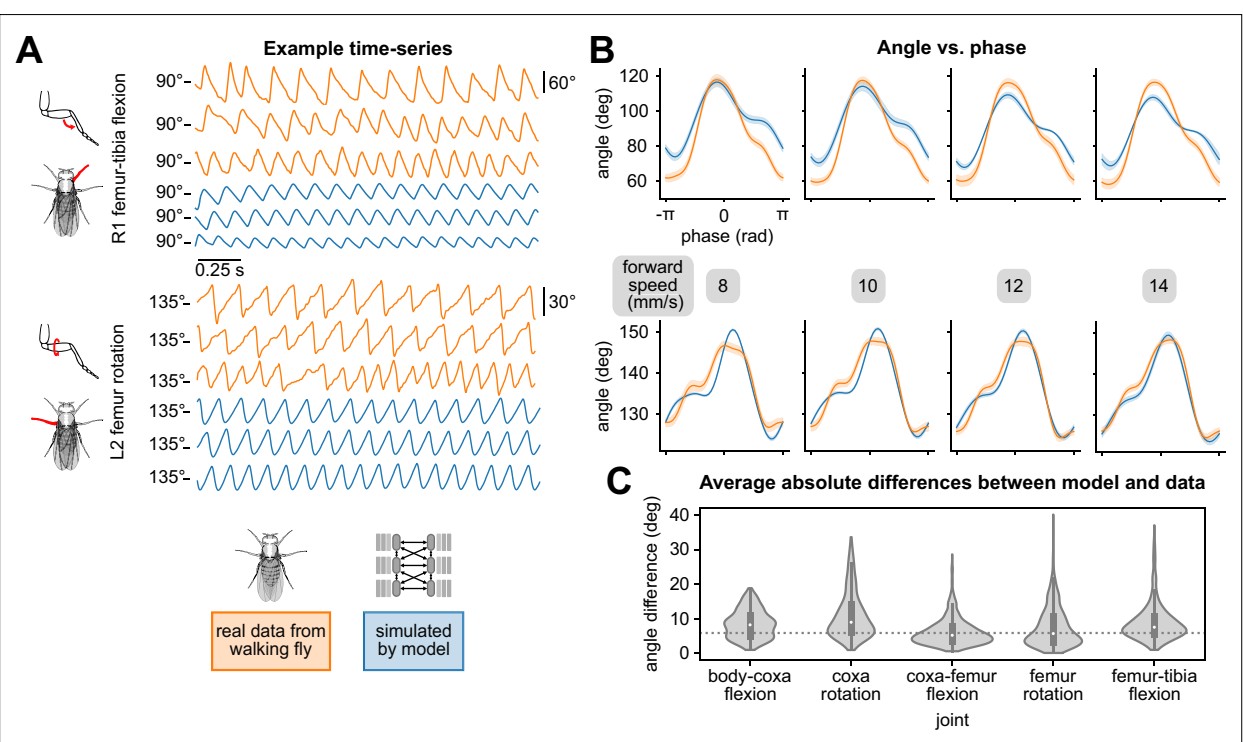

**Figure 2.** Walking model simulations produce realistic joint kinematics. (**A**) Example time series of femur-tibia flexion (**R1**) and femur rotation (**L2**) for three different walking speeds: 8, 10, and 12 mm/s. Real data (orange) exhibited more variability than simulations (blue). (**B**) Angle vs. computed per-leg phase of femur-tibia flexion on leg R1 and femur rotation on leg L2 for four different walking speeds. Each plot contains data from 4 walking bouts with different initial conditions. (**C**) Average differences between model simulations and data, over a range of forward walking, turning, and side-stepping speeds over 500 distinct bouts. The dotted line (5.56 degrees) indicates uncertainty associated with markerless 3D joint tracking (*Karashchuk et al., 2021*). All simulations used a sensory delay of 10 ms and a motor delay of 30 ms, based on values measured experimentally with electrophysiology from leg sensory and motor neurons/muscles (*Tuthill and Wilson, 2016b*; *Azevedo et al., 2020*).



**Video 1.** Model generates forward walking similar to real flies. Shown is an example comparison of real and simulated fly walking kinematics, visualized on a fly model by inverse kinematics (no further physics simulation).

https://elifesciences.org/articles/99005/figures#video1

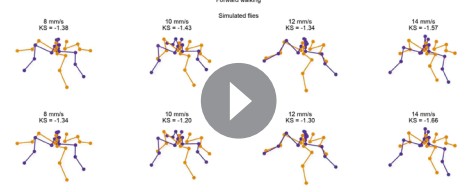

**Video 2.** Model generates forward walking that is visually similar to data. (Top row) Example simulated 3D pose trajectories at 8, 10, 12, and 14 mm/s forward walking (0 mm/s turning and side speeds). (Bottom row) Example 3D pose trajectories from data at the same speeds for comparison. Note that we fix angle joints not included in the model.

https://elifesciences.org/articles/99005/figures#video2

## Control and dynamics

The optimal controller layer maintains walking kinematics in the presence of sensorimotor delays and helps compensate for external perturbations. This design was inspired by optimal control-based models of movements in humans (*Todorov and Jordan, 2002*; *Scott, 2004*; *Berret et al., 2011*). At regular intervals, the controller receives a short time series of desired state trajectories $\theta_d$ from the trajectory generator layer. The controller then produces the necessary torques $\tau$ to track this trajectory for the given leg dynamics. External perturbations $w$, when present, enter through the dynamics and affect the state $\theta, \dot{\theta}$; the controller then senses the state and responds accordingly.

To design the controller, we first derived dynamical equations for each leg using link-and-joint models (*Figure 1C*), then linearized these dynamics and designed a linear quadratic regulator (LQR) controller. This optimal controller senses the state of the leg via proprioceptive input, then determines the optimal motor output for walking. The controller makes use of internal predictive states to accommodate sensory and motor delays. A detailed description of the controller derivations can be found in Methods and materials. We note that the walking and compensation capabilities of the full models are not contingent upon any specific dynamics or controller formulation; any controller that adequately tracks the trajectory generator would suffice.

### The model generates realistic walking kinematics

The layered model generates 3D walking kinematics that resemble real kinematic data from walking flies (*Figure 2*). Below, we provide qualitative and quantitative comparisons of joint angles, joint angular velocities, and phases of walking both within and across legs.

### Qualitative evaluation of joint angle time series and videos of walking kinematics

We first qualitatively compared simulated and real kinematics by examining time series data and videos. In example trajectories of femur-tibia flexion angles of the right front and femur rotation angles of the left middle legs, the simulated time series matched the mean, frequency, and pseudo-triangular shape of the fly data (*Figure 2A*). Articulated animations of simulated and real trajectories are shown in *Videos 1–3*. Although the model and data were largely similar, some differences stood out. For example, the

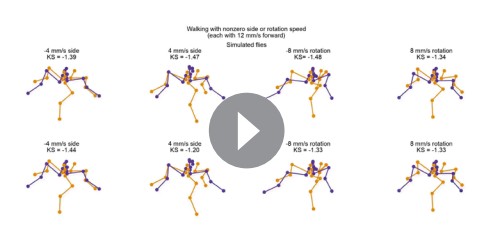

**Video 3.** Model generates walking with nonzero rotation and side speeds visually similar to the data. (Top row) Example simulated 3D pose trajectories of fly walking with some nonzero side or rotation speed. Forward speed is 12 mm/s throughout. (Bottom row) Example 3D pose trajectories from data with similar speeds.

https://elifesciences.org/articles/99005/figures#video3

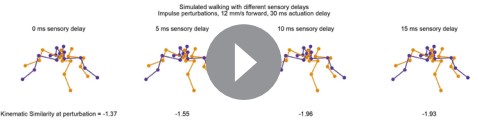

**Video 4.** Model generates robust walking under impulse perturbation with varying motor delays. Shown are example simulated 3D pose trajectories of fly walking with 10 ms sensory delay and varying motor delays. Below are mean kinematic similarity values during the perturbation.

https://elifesciences.org/articles/99005/figures#video4

**Video 5.** Model generates robust walking under persistent stochastic perturbation with varying motor delays. Shown are example simulated 3D pose trajectories of fly walking with 10 ms sensory delay and varying motor delays. Below are mean kinematic similarity values during the perturbation.

https://elifesciences.org/articles/99005/figures#video5

simulated amplitudes were generally smaller than the real amplitudes, and the simulated trajectories tended to be more regular.

## Comparing joint angles and angular velocities versus phase

We next sought to quantitatively compare simulated and real joint angles. Direct comparisons of time series trajectories are inadequate, because temporal offsets between time series produce large mismatches even if the time series are similar. For instance, if we shift a time series trajectory by a half-cycle and compare it with itself, this will produce a large mismatch, even though the two trajectories are identical except for a misalignment in phase.

To compare real and simulated trajectories, we computed the step-cycle phase for each time series and used this to plot the mean angles $\theta$ as a function of phase (*Figure 2B*). Here, we make the distinction between the *generated phase*, the per-leg phases produced by the phase coordinator of the model, and *computed phase*, which can be computed for each joint from time series data. Since we did not have access to the generated, desired phase for real fly data, all comparisons were made between computed phases.

When we averaged joint angles for all legs over 500 distinct walking bouts of 0.5–2 s in duration, we found that the mean differences between real and simulated joint angles were less than 6 degrees (*Figure 2C*). This difference is comparable to the uncertainty associated with markerless tracking of 3D fly walking kinematics that we used as training data (5.56 degrees, from *Karashchuk et al., 2021*). Errors for angular velocity were higher. Aggregate differences as a function of walking and turning velocity are shown in Appendix 4.

The similarity between real and simulated data as a function of phase was consistent across the natural range of forward walking speeds (*Figure 2B*). Plots for all legs and joints are shown in Appendices 3 and 4. We further demonstrate the model's capacity to simulate leg kinematics by comparing the phase coupling of simulated joint kinematics with real walking flies, both within and across legs, in Appendix 5.

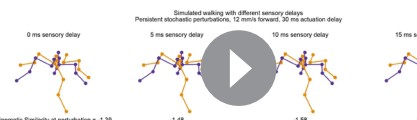

**Video 6.** Model generates robust walking under impulse perturbation with varying sensory delays. Shown are example simulated 3D pose trajectories of fly walking with 30 ms motor delay and varying sensory delays. Below are mean kinematic similarity values during the perturbation.

https://elifesciences.org/articles/99005/figures#video6

**Video 7.** Model generates robust walking under persistent stochastic perturbation with varying sensory delays. Shown are example simulated 3D pose trajectories of fly walking with 30 ms motor delay and varying sensory delays. Below are mean kinematic similarity values during the perturbation.

https://elifesciences.org/articles/99005/figures#video7

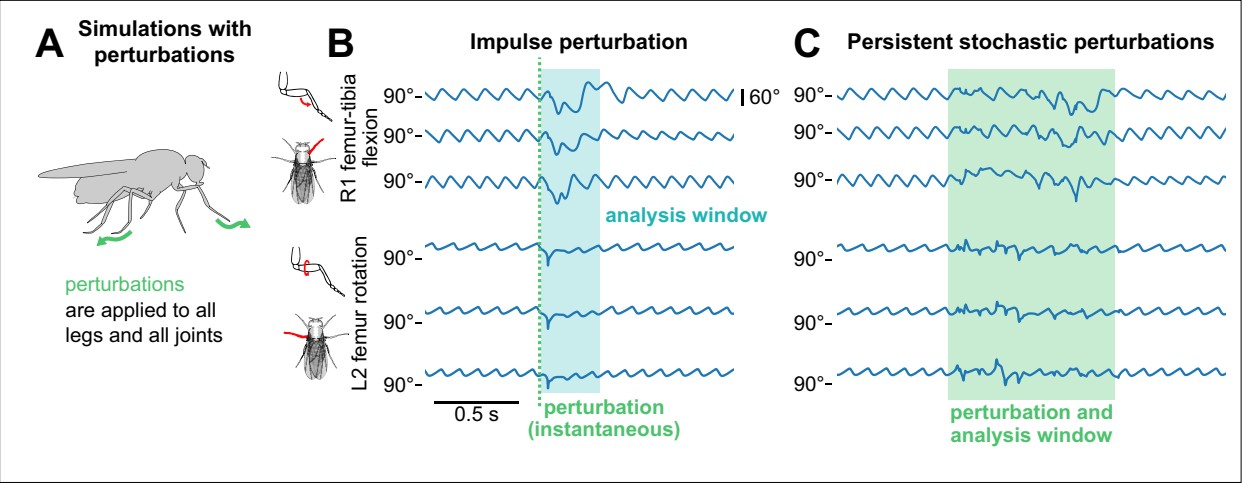

**Figure 3.** The model sustains robust walking during and following perturbations. (**A**) Schematic of how perturbations are applied to the walking model. (**B, C**) Example time series of femur-tibia flexion on leg R1 and femur rotation on leg L2 for three different walking speeds (8, 10, and 12 mm/s) before, during, and after perturbation. In panel B, an impulse perturbation of size 3.75 rad/s is applied at a single instant (dotted line), and its effects are analyzed over a brief time window (shaded blue area). In panel C, stochastic perturbations of size 1.875 rad/s are applied over a time window (shaded green area). Angle trajectories are visibly different during the analysis windows, but recover afterward. Perturbation effects appear similar across speeds. All simulations used a sensory delay of 10 ms and a motor delay of 30 ms.

## Model maintains walking under unpredictable external perturbations

When walking in natural environments, animals frequently navigate uneven or slippery terrain. Thus, robust sensorimotor control systems must detect and respond to such unexpected perturbations in order to maintain stable locomotion.

Here, we show that our model maintains realistic walking in the presence of external dynamic perturbations, despite being trained only on data of walking without perturbations (no perturbation data was available). This performance is made possible by the combination of the trajectory generator and the optimal controller in the model. Taking advantage of proprioceptive feedback, the optimal controller compensates for external perturbations, allowing the trajectory generator to sustain realistic joint angle trajectories.

We considered two types of perturbations: impulse and persistent stochastic. Impulse perturbations are analogous to when an animal experiences a brief, unexpected motion (e.g. legs slipping on an unstable surface). We simulated impulse perturbations as a velocity that is added to all joints at a single time (i.e. a single time step in our simulation). The magnitude of this velocity was drawn from a normal distribution for each joint, where increasing perturbation strength increases the mean and variance of this distribution. The sign of the velocity was drawn separately so that there is an equal likelihood for negative or positive perturbation velocities. Persistent stochastic perturbations displaced all legs, but not at the same time. We applied perturbations following a Poisson process with a mean rate of 10 Hz; in other words, over a period of time (1 s), for each leg at each timestep, we randomly selected whether an impulse perturbation should be applied based on a Poisson distribution. Ranges of perturbation strengths used in our simulation correspond to estimates of biologically plausible values (as derived in Appendix 6).

To evaluate the model's ability to walk in the presence of perturbations, we compared time series data before, during, and after perturbations across a range of walking speeds (*Figure 3B–C*). For both types of perturbation, joint angle trajectories were different during perturbations, but recovered after the perturbation period ended. In the case of persistent stochastic perturbations, kinematics appeared approximately oscillatory even during perturbations, indicating that a semblance of walking was maintained.

Example animations of simulated walking bouts with perturbations are shown in *Videos 4–7*.

To quantify the extent to which perturbed kinematics resemble normal, unperturbed walking, we introduce a new quantitative metric termed *kinematic similarity (KS)*. For a given window of a kinematic trajectory, KS is computed by the log-likelihood that it occurred in the real fly-walking data

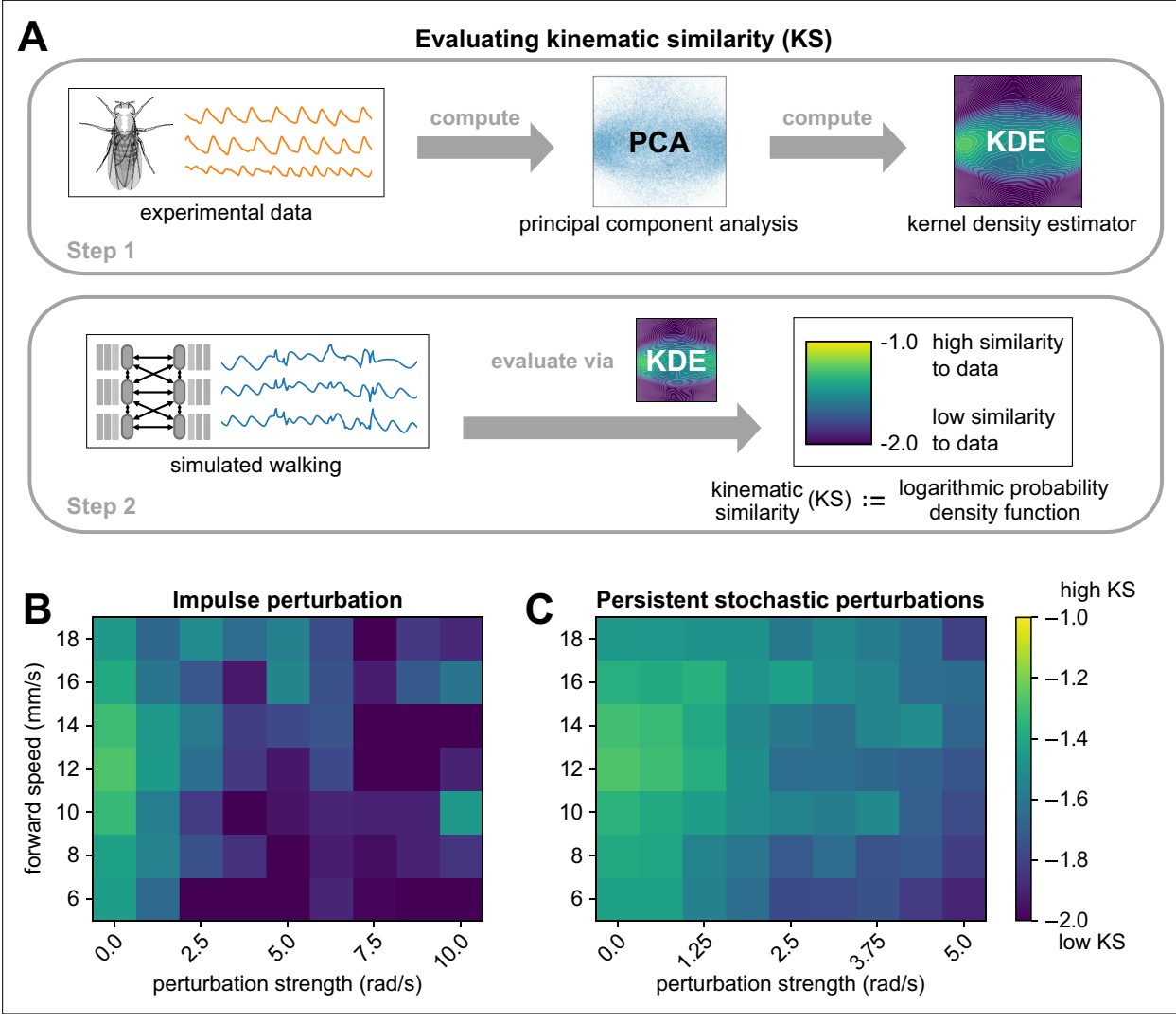

**Figure 4.** The walking model sustains realistic walking kinematics following impulse and persistent stochastic perturbations. (**A**) Method for computing kinematic similarity between real data and simulated walking. First, the full set of experimental data (from flies) is used to compute a Gaussian kernel density estimator (KDE). To quantify the similarity (to data) of a given bout of simulated walking, we apply the KDE to evaluate the log probability density function of each bout, a scalar value we refer to as kinematic similarity (KS). High KS indicates that the perturbed walking resembles the unperturbed walking from data, while low KS indicates that the perturbed walking deviates from data and results in unnatural angles (as seen at 40 ms motor delay). The average KS value of bouts from experimental data was –1.627. (**B, C**) Kinematic similarity of simulated walking during perturbations for impulse and persistent stochastic perturbations. For each square of the heatmap, four simulations with different initial conditions were simulated and averaged. Simulations became less similar to data with increased perturbation strength. All simulations used a sensory delay of 10 ms and a motor delay of 30 ms. Perturbation velocities were drawn from a random uniform distribution with mean *perturbation strength* (rad/s) and standard deviation of 0.1 × mean.

(illustrated in *Figure 4A* and detailed in Methods and materials). Briefly, we reduce the experimental data to 2 dimensions using principal components analysis (PCA), then fit a kernel density estimator (KDE) to the resulting distribution. We then project each bout of simulated walking onto this subspace and evaluate the KDE model to obtain a log probability density function estimate, which corresponds to the kinematic similarity of the simulated bout to real walking bouts in the data. Lower values of KS mean lower similarity to data. When we applied this method to bouts from experimental data, we found that the average KS of experimental data was –1.627. Thus, we use KS $> -1.6$ as a general threshold for evaluating the realism of joint angle trajectories.

As expected, larger perturbations led to walking behaviors with lower KS, and KS was not strongly dependent on forward walking speed (*Figure 4B–C*). For persistent stochastic perturbations, the model produced realistic joint angle trajectories (KS $> -1.6$) for all simulated perturbation strengths.

Impulse perturbations appeared to be more challenging for the model, as they resulted in lower KS. From the time series, we observed that impulse perturbations resulted in greater instantaneous deviation from the standard waveform than persistent stochastic perturbations of the same magnitude. This is due to the fact that impulse perturbations produce simultaneous changes to all legs and joints, whereas persistent stochastic perturbations are spread out in time and typically non-simultaneous.

### Effect of sensory and motor delays on walking

Temporal delays are inherent properties of sensorimotor control systems, but they are difficult to manipulate experimentally (*Bässler, 1993*). Therefore, we used our model to investigate how changing sensory and motor delays affect locomotor robustness. We used measurements from the literature to estimate physiological delays for leg sensory and motor neurons in the *Drosophila* leg. Our estimate of sensory delay (5–15 ms) was based on the measured delay from spike initiation in a mechanosensory neuron in the *Drosophila* femur to the peak of an excitatory postsynaptic potential in a postsynaptic VNC neuron (*Tuthill and Wilson, 2016b*). Our estimate of motor delay (20–40 ms) was based on the time between spike initiation in a tibia motor neuron cell body to the onset of muscle force production, measured with a force probe (*Azevedo et al., 2020*).

Without external perturbations, the model produced realistic walking with arbitrary delays, since the controller can effectively compensate for large delays by using predictions of joint angles in the future. However, in the presence of external perturbations and high delay values, the model was unable to maintain realistic walking, since it could not respond rapidly enough to unexpected perturbations.

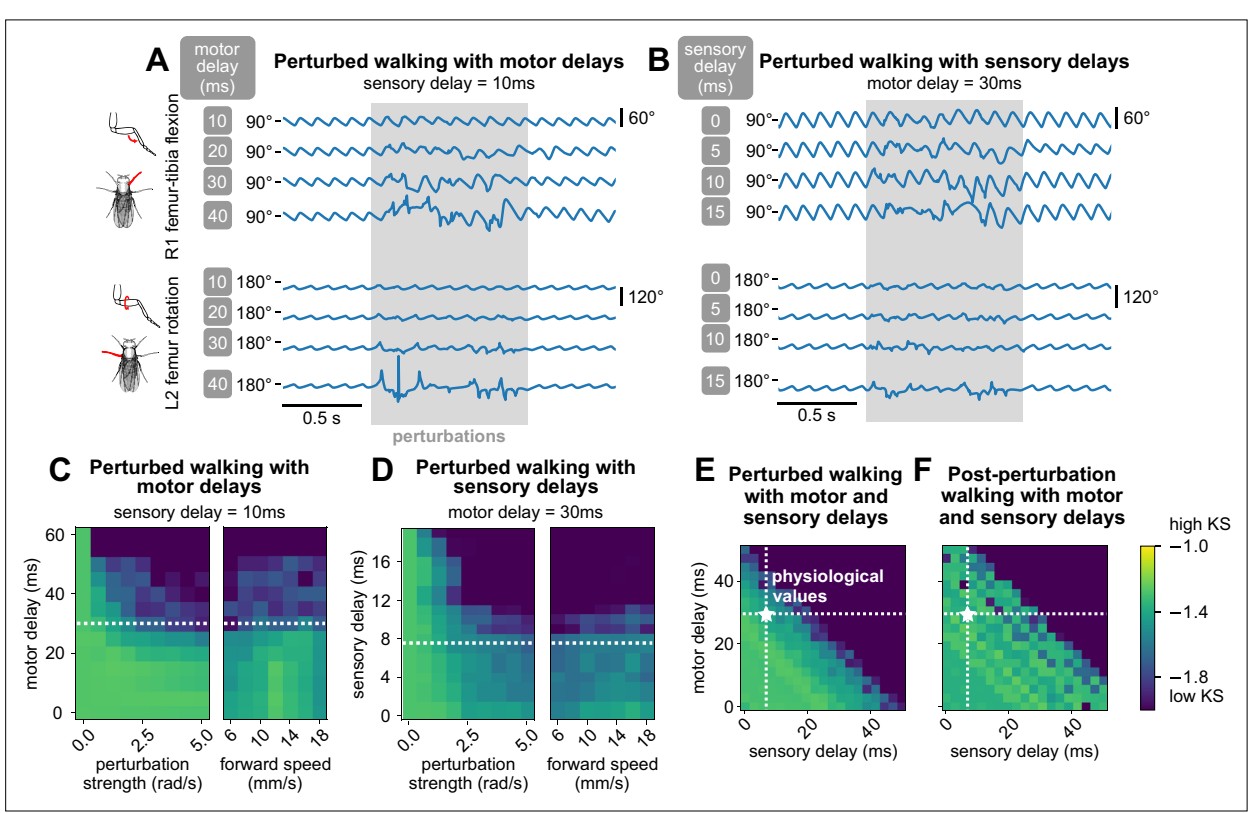

**Figure 5.** The model maintains robust walking under persistent stochastic perturbations over specific ranges of motor and sensory delays. (**A, B**) Example time series of femur-tibia flexion on leg R1 and femur rotation on leg L2 under various values of motor (10, 20, 30, 40 ms) and sensory delay (0, 5, 10, 15 ms). Perturbation effects became more noticeable with increasing delay values. (**C, D, E**) Similarity of during-perturbation walking to data across delay values, perturbation strengths, and forward speeds. For each square of the heatmap, four simulations with different initial conditions were simulated and evaluated. At low delay values, simulated walking maintained high similarity even under large perturbations. As perturbation strength and delays increased, simulated walking became less similar to data; the effect was more pronounced with increased delays. When we fixed one delay value and varied the other, the model maintained realistic walking (KS > – 1.6) up to about 30 ms of motor delay and 10 ms of sensory delay across a range of conditions. When we allowed both motor and sensory delay to vary, the model maintained realistic walking when the sum of the delays was no more than about 45 ms. (**F**) Post-perturbation walking with motor and sensory delays. The model was unable to recover from perturbations for large delay values. Unless otherwise stated, forward speed = 12 mm/s, and perturbation strength = 1.875 rad/s.

Here, we consider the composite effects of persistent stochastic perturbations with motor and sensor delays. Similar results for impulse perturbations are included in Appendix 7.

We first fixed the sensory delay at 10 ms and measured the effect of varying motor delays. Examining the time series data, we found that for low values of motor delay (10 ms, 20 ms), even stronger perturbations had almost no effect (*Figure 5A*). However, at higher values of motor delay, the effects of the same perturbation became more pronounced, although the model still managed to recover after the perturbations ended. Over a range of perturbation strengths and walking speeds, the model maintained realistic walking (KS $> -1.6$) up to about 30 ms of motor delay (*Figure 5C*). This value is consistent with motor delays measured from the fruit fly leg, where the time between motor neuron spiking to the onset of muscle force production is about 30 ms (*Azevedo et al., 2020*).

Next, we fixed the motor delay at 30 ms and observed the effect of varying sensory delays. From time series data, we found that even at low values of sensory delay (0 ms, 5 ms), the effects of perturbations on model output were significant (*Figure 5B*). When sensory delay increased, the effects of the perturbation became more pronounced. Over a range of perturbation strengths and walking speeds, the model maintained realistic walking (KS $> -1.6$) up to about 10 ms of sensory delay (*Figure 5D*). This range is consistent with experimental estimates of delays from mechanosensory neurons in the fly femur (*Tuthill and Wilson, 2016b*), although delays could be even longer for more distal sensors. For example, a sensory neuron at the tip of the tarsus would take ~20 ms to travel 2 mm at an estimated conduction velocity of ~0.3 m/s (*Tuthill and Wilson, 2016b*).

For any fixed set of values of motor and sensory delays, the KS of perturbed walking was not dependent on forward walking speed. In other words, slower walking did not improve the model's ability to sustain walking during perturbations (*Figure 5C and D*). We expand on this observation in the Discussion.

Last, we explored combinations of values for motor and sensory delays for fixed walking speed and perturbation strength. We observed that robust walking was not contingent on the specific values of motor and sensory delay, but rather the *sum* of these two values (*Figure 5E*). Furthermore, as total delay increased, higher frequencies of walking were impacted first before walking collapsed entirely (Appendix 9). Overall, in order for the model to overcome a perturbation, the key parameter was the total delay between the perturbation onset and the motor response.

In summary, we used a virtual fly model of walking to perform in silico manipulations of sensory and motor delays, including delays exceeding physiologically-realistic values. When we tested the model across a range of delay values, we found that the model was able to maintain robust walking only within the physiological range. Outside of this range, the model could not maintain walking that was kinematically similar to real flies. This finding suggests that the fly locomotor control system operates close to the temporal limit at which it can detect and respond to external perturbations. Thus, we propose that sensory and motor delays establish a fundamental tradeoff between the speed and robustness of fly walking.

## Discussion

In this paper, we develop a computational model that realistically imitates the 3D joint kinematics of walking *Drosophila* and incorporates sensorimotor delay as a tunable parameter. We then used the model to establish a quantitative relationship between sensorimotor delays and locomotor robustness. We found that the model's ability to maintain walking following external perturbations significantly degrades for delay values that exceed known physiological values, suggesting that these parameters are fundamental constraints on fly locomotion. The formulation of a modular, multi-layered model for locomotor control makes new experimentally testable hypotheses about fly motor control and can also be applied to investigate limbed locomotion in other organisms. Future extensions of the model to improve its realism may include premotor neural circuits from the fly connectome (*Lesser et al., 2024*; *Cheong et al., 2024*) and biomechanical interactions between the limb and the environment (*Lobato-Rios et al., 2022*; *Wang-Chen et al., 2023*; *Vaxenburg et al., 2024*).

# Fundamental constraints on locomotion imposed by sensory and motor delays

Sensory and motor delays are inextricable properties of animal locomotor systems. To study the impact of delays on locomotor control, we developed a hierarchical walking model with explicit inclusion of physiological delays as tunable parameters. Importantly, the model incorporates delay while preserving behavioral realism, laying the groundwork for future studies on the effect of delay on other aspects of locomotion and sensorimotor control.

Our model predicts that at the same perturbation magnitude, walking robustness decreases as delays increase. This could be experimentally tested by altering conduction velocities on the fly, for example by increasing or decreasing the ambient temperature (*Banerjee et al., 2021*). If a warmer ambient temperature decreases delays in the fly, but fly walking robustness remains the same in response to a fixed perturbation, this would indicate a stronger role for central control in walking than our modeling results suggest.

Trade-offs between energy, performance, and robustness establish fundamental constraints on animal locomotion. It is energetically expensive to build and maintain muscles and neurons that operate with low delay (*Sterling, 2015*). From a performance perspective, long delays limit reaction speed and control robustness, which can ultimately impact animal survival (*More and Donelan, 2018*). A third consideration is locomotor robustness, or the capacity to detect and respond to external perturbations. In flies, the axons of leg motor neurons and some proprioceptor axons are among the largest diameter cables in the leg nerve (*Phelps et al., 2021*), suggesting that the speed of these systems is under selective pressure.

Our model suggests that fly walking operates in a middle ground with respect to speed and robustness. Specifically, our model maintains normal walking for up to about 30 ms of motor delay and 10 ms of sensory delay. These values are strikingly close to measured values from *Drosophila*: 30 ms from motor neuron spike to peak force production in a femur muscle (*Azevedo et al., 2020*) and 6ms from a femoral mechanosensory neuron spike to the onset of a postsynaptic response in the VNC (*Tuthill and Wilson, 2016b*).

In our model, robust locomotion was constrained by the cumulative sensorimotor delay. This result could be experimentally validated by comparing how animals with different ratios of sensory to motor delays respond to perturbations. Alternatively, it may be possible to manipulate sensory vs. motor delays in a single animal, perhaps by altering the development of specific neurons or ensheathing glia (*Kottmeier et al., 2020*). If sensory and motor delays have significantly different effects on walking quality, then additional compensatory mechanisms for delays could play a larger role than we expect, such as prediction through sensory integration, mechanical feedback, or compensation through central control. A rich and related topic for future exploration is the interaction of delays with body size and behavioral ecology. Longer limbs, heavier bodies, and more mechanically complex tasks may alter the tradeoffs between energetics, speed of force generation, noise, and robust sensorimotor control (*Labonte et al., 2024*; *Sutton et al., 2023*).

## Role of proprioceptive feedback in fly walking

Our model provides insight into the role of proprioceptive feedback in fly walking, which remains an active area of research (*Dallmann et al., 2021*). Many models of fly walking ignore the role of feedback, relying instead on central pattern generators (*Lobato-Rios et al., 2022*; *Szczecinski et al., 2018*; *Aminzare et al., 2018*) or metachondral waves (*DeAngelis et al., 2019*) to model kinematics. Some models incorporate proprioceptive feedback, primarily as a mechanism that alters timing of movements in inter-leg coordination (*Goldsmith et al., 2020*; *Wang-Chen et al., 2023*). Experimental work in cockroaches suggests that these fast-running insects rely on central control and mechanical feedback, particularly at high speeds (*Couzin-Fuchs et al., 2015*; *Ayali et al., 2015b*). In contrast, studies in stick insects have shown that these slow-walking animals are highly dependent on proprioceptive feedback for leg coordination during walking (*Bässler, 1977*; *Ayali et al., 2015a*; *Schilling et al., 2013*).

Silencing mechanosensory chordotonal neurons alters step kinematics in walking *Drosophila* (*Mendes et al., 2013*; *Pratt et al., 2024*). Additionally, removing proprioceptive signals via amputation interferes with inter-leg coordination in flies at low walking speeds (*Berendes et al., 2016*). However, the role of leg proprioception in overcoming external perturbations has not previously been

studied in flies. In our model, which does not include limb compliance or other biomechanical adaptations, the fly effectively overcomes perturbations using only proprioceptive feedback. The need for proprioception to compensate for perturbations may be reduced when biomechanical mechanisms are present. Nonetheless, we hypothesize that removing proprioceptive feedback would impair an insect's ability to sustain locomotion following external perturbations.

## Predictive control is critical for responding to perturbations due to motor delay

Sustaining realistic walking kinematics in the presence of perturbations and motor delays is a challenging task for any model. In our model, this design criterion motivated us to develop a compensatory prediction in the optimal controller formulation, based on *Stenberg et al., 2022*. The controller compensates for the known motor delay by predicting future dynamics. We found that the model was quite sensitive to the *prediction horizon*, i.e., how far into the future the controller predicts. The model works best when the controller's prediction horizon matches the motor delay — we thus used matched values in all simulations. We experimented with altering the prediction horizon to be less than the motor delay, with catastrophic consequences: the model ceased walking and mostly produced noise (not shown). From this observation, we deduce that future predictions are crucial in compensating for motor delays in our walking model, in agreement with previous theoretical work on predictive or 'forward' models for sensorimotor control (*Desmurget and Grafton, 2000*; *Li et al., 2023*). We propose that fly motor circuits may encode predictions of future joint positions, so the fly may generate motor commands that account for motor neuron and muscle delays. Consistent with this hypothesis, *Dallmann et al., 2023* recently found that descending motor commands from the brain excite GABAergic interneurons in the VNC that inhibit velocity-encoding proprioceptors. Thus, some proprioceptive feedback signals from the fly leg are predictively suppressed during self-generated movements.

## Layered model produces robust walking and facilitates local control

One key finding of our model is that robust walking in the presence of external perturbations can emerge from a local controller in combination with a trajectory generator that is trained only on perturbation-free walking data. We did not model adaptation because compensation emerges as a property of a tuned controller with feedback. The success of this simple, layered model suggests that fast, robust locomotion could be maintained locally, requiring minimal plasticity within central circuits.

Layering is a familiar concept in network architecture and hardware/software stacks (*Chiang et al., 2007*) and has recently found applications as a modeling framework in biology and neuroscience (*Doyle and Csete, 2011*; *Nakahira et al., 2021*). In our model, the complex problem of multi-legged locomotion is broken down into three sub-problems (delayed control, joint kinematic generation, inter-leg coordination), each of which is delegated to a different layer.

Each layer provides an abstracted interface for the layer below it, which can also be thought of as a special case of modularity. The separation of function between layers reinforces that robust locomotion can be produced by local (i.e. per-leg) control signals. Per-leg modules could also be extended to control more legs, generalizing to locomotor control in isopods and millipedes. Instead of redesigning the overall coordination and locomotor control circuits, basic duplication and subsequent fine-tuning may be sufficient.

Beyond walking, our layered model framework could be applied to other animals with other locomotor strategies, including flying, swimming, slithering, and digging. The key ingredients required for the model are: (1) a functional inter-limb coordinator, (2) sufficient 3D kinematics data to train a trajectory-generator layer, and (3) a controller that adequately tracks the trajectory for some dynamical models of the animal. The dynamical model may be linearized (as we did here) and controlled with a standard controller; other approaches include successive (per-timestep) linearization with linear model predictive control (MPC) (*Berberich et al., 2022*), or fully nonlinear control techniques (*Slotine and Li, 1991*).

The layered model approach also has potential applications for biomimetic robotic locomotion. For simplicity, we used a linearized link-and-joint dynamical model. However, one could also replace this dynamical model to control a hexapod robot such as in *Goldsmith, 2019*. Due to its modular nature, the other modules (trajectory generator, inter-leg coordinator) would remain unchanged and

the resulting model could, in theory, generate 3D kinematics for bio-mimetic walking on a robotic platform. Overall, the model can be thought of as performing a layered implementation of imitation learning, which is a popular technique in robotics (*Hua et al., 2021*; *Johns, 2021*).

## Towards biomechanical and neural realism

The goal of our model was to produce realistic 3D joint kinematics while incorporating sensory and motor delays. To achieve this, the model contains several physiological simplifications. First, our dynamics model did not allow dynamical coupling between legs through the mechanics of the body, as the legs are only coupled neurally through the phase coordinator. However, in real bodies, the legs are also dynamically coupled through the body and its weight distribution over the legs (*Dallmann et al., 2017*). Our model also did not consider explicit leg-ground contact interactions. Rather, interactions with the ground were implicitly taken into account by the trajectories learned by the neural network, though they were not made explicit in the dynamics. Our goal was to mimic the kinematic trajectory, a problem known in robotics as *motion control*. However, including ground contact interactions would require computing ground contact forces, which are currently unavailable in the kinematics dataset we used.

In order to model ground contact forces and joint torques, force-based learning would need to be incorporated into the trajectory generator, requiring a new dataset or extrapolating force values (e.g. via a physics-based model). Furthermore, the controller would need to be reformulated to use *impedance control* or *hybrid control* techniques (*Buss et al., 2002*; *Arevalo and Garcia, 2012*; *Sciavicco and Siciliano, 2012*). These approaches are common in robotics to deal with dynamics control problems, which concern both kinematic trajectories *and* external contact forces. The inclusion of explicit leg-ground contact interactions would also make it harder for the model to recover when perturbed, because perturbations during walking often occur upon contact with the ground (e.g. the ground is slippery or bumpy).

A promising avenue for future investigation is an integration of our controller architecture with a virtual physics model (*Lobato-Rios et al., 2022*; *Wang-Chen et al., 2023*; *Vaxenburg et al., 2024*), which would facilitate the incorporation of dynamical coupling between legs, as well as leg-ground contact interactions. The inclusion of these features may require additional coordination between the legs, which might decrease allowable values of sensory and motor delay.

Another step toward biological realism is the incorporation of explicit dynamical models of proprioceptors, muscles, tendons, and other biomechanical aspects of the exoskeleton. The proprioceptive neurons in the femoral chordotonal organ of each fly leg encode angles and angular derivatives (*Mamiya et al., 2018*). Additional proprioceptive feedback is provided by hair plate sensory neurons (limit detectors) and campaniform sensilla (load sensors), which are distributed across each leg (*Tuthill and Wilson, 2016a*). Thus, our use of joint state $\theta, \dot{\theta}$, likely underestimates the resolution of proprioceptive feedback to the fly motor system. We anticipate that the increased sensory resolution from more detailed proprioceptor models and the stability from mechanical compliance of limbs in a more detailed biomechanical model would make the system easier to control and increase the allowable range of delay parameters. Conversely, we expect that modeling the nonlinearity and noise inherent to biological sensors and actuators may decrease the allowable range of delay parameters. In the stick insect, load-sensing campaniform sensilla appear to have greater conduction delays than movement-sensing proprioceptors in the femoral chordotonal organ (*Gebehart and Büschges, 2021*). Future models may investigate how these different delays from different proprioceptive sensors impact sensorimotor control.

A further step towards neural realism would be to constrain the trajectory generator and optimal controller using patterns of synaptic connectivity within sensorimotor circuits of the fly VNC. This is now feasible using recent connectomes of the *Drosophila* VNC (*Azevedo et al., 2024*; *Takemura, 2023*). However, many challenges remain for connectome-constrained models, because many important physiological parameters are still unknown.

Future work may also model how higher-level planning of fly behavior interacts with the lower-level coordination of joint angles and legs. Walking flies continuously change their direction and speed as they navigate the environment (*Katsov et al., 2017*; *Iwasaki et al., 2024*). Past work shows that flies tend to recover and walk at similar speeds following perturbations (*DeAngelis et al., 2019*), but individual flies might still change walking speed, phase coupling, or even transition to other behaviors,

such as grooming. Modeling these higher-level changes in behavior would involve combining our sensorimotor model with models for navigation (*Fisher, 2022*) and behavioral transitions (*Berman et al., 2016*).

Although we believe our model matches the fly walking sufficiently for this investigation, we do note that our model still underfits the joint angle oscillations in the walking cycle of the fly (see *Figure 2* and Appendix 2). More precise fitting of the joint angle kinematics may come from increasing the complexity of the neural network architecture, improving the training procedure based on advances in imitation learning (*Hussein et al., 2018*), or explicitly accounting for individual differences in kinematics across flies (*DeAngelis et al., 2019*; *Pratt et al., 2024*).

Our layered approach could be a useful framework for learning principles from closed-loop models, despite having incomplete information about the biological system. For instance, the trajectory generator model could be replaced by a combination of a connectome simulation and artificial neural networks, with the constraint on the artificial neural network dynamics coming from the complete circuit generating robust walking in simulation. Future work could also incorporate more detailed delay structures based on compartmental models of neurons from the connectome. With increasing realism, such closed-loop models provide a promising means to investigate how complex neural circuits interact with proprioceptive feedback to control robust locomotor behaviors.

## Materials and methods

### Tracking joint angles of *D. melanogaster* walking in 3D

We obtained fruit fly *D. melanogaster* walking kinematics data following the procedure previously described in *Karashchuk et al., 2021*. Briefly, a fly was tethered to a tungsten wire and positioned on a frictionless spherical treadmill ball suspended on compressed air. Six cameras (Basler acA800-510 µm with Computar zoom lens MLM3X-MP) captured the movement of all of the fly's legs at 300 Hz. The fly size in pixels ranges from about 300×300 up to 700×500 pixels across the six cameras. Using Anipose, we tracked 30 keypoints on the fly, which are the following five points on each of the six legs: body-coxa, coxa-femur, femur-tibia, and tibia-tarsus joints, as well as the tip of the tarsus.

To fit the model described in this paper, we extracted a subset of the tracking data when the fly walking, as opposed to non-walking behaviors including standing, grooming, etc. To isolate bouts of walking, we used the behavior classifier described in *Karashchuk et al., 2021*. We further selected walking bouts of at least 0.5 s (150 video frames) in length, and where the femur-tibia flexion angle of the left front leg had a range of at least 30 degrees.

In total, our dataset consisted of 3473 walking bouts from 45 flies total. The average length of a walking bout was 0.877 s (263 frames), with 3049.7 s of walking total (914,909 frames).

### Inter-leg phase coordinator

We model the coordination between legs as phase-coupled Kuramoto oscillators (*Strogatz, 2000*), where the frequency of each oscillator is driven by the trajectory generator described in the next section.

Specifically, the phase for a leg $i$ is $\phi_i$ and evolves according to its derivative $\dot{\phi}_i$

$$\dot{\phi}_i = F_i(\theta_i, \dot{\theta}_i, v, \phi_i) + \alpha \sum_{j \neq i} \sin(\phi_j - \phi_i - \bar{\phi}_{ij}), \tag{1}$$

where $F_i$ is the trajectory generator function for leg $i$, $\alpha$ is the coupling strength, and $\bar{\phi}_{ij}$ is the steady-state phase offset between legs $i$ and $j$.

We model the coupling across the legs as all-to-all coupling, with coupling strength $\alpha = 6.5$. We found this coupling strength best reproduced the phase coupling distributions from the real data (as shown in Appendix 5).

We estimate $\bar{\phi}_{ij}$ from the walking data by taking the circular mean over phase differences of pairs of legs during walking bouts. We find that the phase offset across legs is not strongly modulated across walking speeds in our dataset (see Appendix 1), so we model $\bar{\phi}_{ij}$ as a single constant independent of speed. In future studies, this could be a function of forward and rotation speeds to account for fine phase modulation differences.

## Trajectory generator

A trajectory generator model was formulated for each of the six legs and fit separately to fly walking data tracked during tethered walking without any external perturbations.

## Model formulation

We formulate the trajectory generator as the function

$$(\ddot{\theta}, \dot{\phi}) = F(\theta, \dot{\theta}, v, \phi), \tag{2}$$

where $\theta$ is a vector of joint angles, $\dot{\theta}$ is a vector of joint angle derivatives, $v$ is the desired walking speed and direction, and $\phi$ is the phase of the leg. Initially, we explored using the trajectory generator to directly output angles and angular velocities $\theta, \dot{\theta}$; however, we found that more realistic (i.e. similar to data) trajectories were produced when we used the trajectory generator to output angular acceleration $\ddot{\theta}$, which we integrated to produce the desired angle and angular velocities. Note that $v$ is not communicated to or from the layers above and below; instead, we consider walking speed and direction to be given as commands descending from the brain.

To compute a trajectory given an input $v$ and an initial $\phi$, $\theta$, and $\dot{\theta}$, we integrate the function $F$ numerically using the midpoint method (*Lotkin, 1956*). Following methods from *Holden et al., 2017* and *Zhang et al., 2018*, we represent the function $F$ as a multi-layer perceptron neural network with 2 hidden layers of 512 units each. We use ELU (*Clevert et al., 2015*) as our nonlinearity. In total, the multi-layer perceptron has 274,437 parameters for T1 legs and 272,388 parameters for T2 and T3 legs, with the slight difference in parameters due to the different number of joint angles (dimension of $\theta$) modeled for a given leg.

## Training data

To train the multi-layer perceptron network used to represent $F$, we used the fly walking data, tracked as described in the section above. The training data consists of joint angles $\theta$, computed $\dot{\theta}$ and $\ddot{\theta}$, and walking velocity $v$. We estimated the walking cycle phase $\phi$ using a Hilbert transform over the femur-tibia flexion angle for T1 legs, femur rotation angle for T2 legs, and coxa-femur angle for T3 legs. For each phase, we filtered the corresponding angle using a first-order Butterworth bandpass filter with 3 Hz and 60 Hz as critical frequencies, using the scipy library (*Virtanen et al., 2020*). Then, we applied a Hilbert transform to each angle to obtain a complex waveform. We estimate the walking cycle phase from each complex waveform by estimating the angle of each point in the waveform.

## Training procedure

Training the neural network representing $F$ from data was performed in two steps, minimizing its error in predicting one time step, then minimizing its error in predicting a short time trajectory.

In the first step, we minimized the error of $F$ for predicting $(\ddot{\theta}, \dot{\phi})$ over one time step, given the corresponding $(\theta, \dot{\theta}, v, \phi)$ from the training data. We minimized the mean squared error of the prediction, normalized by the variance for each dimension. We trained our network for 300 iterations over the full training data using a batch size of 2500 training samples, using gradient descent with the Adam algorithm (*Kingma and Ba, 2017*). To ensure a robust function at this step, we applied dropout to a random 5% of the hidden units (*Srivastava et al., 2014*). We standardized the input and output training data to the multi-layer perceptron so that it has a mean of 0 and standard deviation of 1.

In the second step, we minimize the error of $F$ for predicting a trajectory $\theta$ when numerically integrated over a short time horizon in the future. Specifically, we integrate $F$ over $T = 60$ steps given initial conditions $(\theta, \dot{\theta}, v, \phi)$ to produce an estimated desired trajectory of $\theta_d(t)$. Here, we minimized the loss:

$$\sum_{t}^{t+T} \|\cos(\theta_d(t)) - \cos(\theta(t))\|_2^2 + \|\sin(\theta_d(t)) - \sin(\theta(t))\|_2^2 \tag{3}$$

using gradient descent with the Adam algorithm (*Kingma and Ba, 2017*). During training, we clip gradients to a norm of 10 to stabilize training.

The training was implemented using Tensorflow (*Abadi, 2015*) running on a computer with NVIDIA GeForce RTX 2070 GPU and AMD Ryzen Threadripper 1920X 12-Core Processor.

**Table 1.** Joints included for leg models.

| Joint | Front legs | Middle legs | Hind legs |
|---|---|---|---|
| Body-coxa flexion | ✓ | | |
| Coxa rotation | ✓ | | |
| Coxa-femur flexion | ✓ | ✓ | ✓ |
| Femur rotation | | ✓ | ✓ |
| Femur-tibia flexion | ✓ | ✓ | ✓ |

## Leg dynamics and optimal controller formulation

All techniques used for dynamics formulation are standard tools from control theory. We begin with a link-and-joint model of the fly leg, as shown in *Figure 1C*. For simplicity, we only model joints that exhibit large ranges of movement during naturalistic walking and turning. For instance, varying femur rotation is important to the movements of the middle legs, but the front legs exhibit near-constant femur rotation *Karashchuk et al., 2021*; thus, a femur rotation joint is included for the middle and hind legs only. The joints included for each leg are shown in *Table 1*.

We write the Denavit-Hartenberg (DH) table of the leg model and use this to systematically derive the Euler-Lagrange matrix equations of motion:

$$\tau = M(\theta)\ddot{\theta} + C(\theta, \dot{\theta})\dot{\theta} + B(\theta)\dot{\theta} + g(\theta),$$ (4)

where $\tau$ is the vector of joint torques; $\theta$, $\dot{\theta}$, and $\ddot{\theta}$ are vectors of joint angles, angular velocity, and angular acceleration; $M$, $C$, $B$, are the inertia, Coriolis, and friction matrices, and $g$ is the gravity vector.

Let us define the state to be the angles and angular derivatives $q = \begin{bmatrix} q_1 \\ q_2 \end{bmatrix} = \begin{bmatrix} \theta \\ \dot{\theta} \end{bmatrix}$ and input to be torques $\tau$. We next rearrange (4) into the form $\dot{q} = F(q, \tau)$, so that

$$\begin{bmatrix} \dot{q}_1 \\ \dot{q}_2 \end{bmatrix} = \begin{bmatrix} q_2 \\ -M(q_1)^{-1} \left( C(q_1, q_2)q_2 + B(q_1)q_2 + g(q_1) \right) \end{bmatrix} + \begin{bmatrix} 0 \\ M(q_1)^{-1} \end{bmatrix} \tau.$$ (5)

To linearize this system, we choose equilibrium values $\bar{q}$ and $\bar{\tau}$, such that $F(\bar{q}, \bar{\tau}) = 0$. In particular, $\bar{q}_1$ is taken to be the average joint angles for each leg joint computed from the data. It follows that $\bar{q}_2 = 0$, and $\bar{\tau} = g(\bar{q}_1)$ gives the desired equilibrium.

We linearized about this equilibrium point, which leads to an equation of the following form:

$$\dot{x} = A_c x + B_c u,$$ (6)

where $x = q - \bar{q}$ and $B_c$ are the Jacobians with respect to $q$ and $\tau$, respectively. In other words,

$$A_c = \frac{\partial F}{\partial q}(\bar{q}, \bar{\tau})$$ (7)

$$B_c = \frac{\partial F}{\partial \tau}(\bar{q}, \bar{\tau}).$$ (8)

In our code, we use the SymPyBotics toolbox (*Sousa, 2013*) to obtain symbolic equations for the quantities in *Equation 4*, then numerically compute Jacobian values.

Next, we rewrite the system in discrete time using a sampling interval $T$, which is typically chosen to be an integer multiple of the sampling interval from the data (the tracking data was acquired every $T = 1/300$ s). In our controller simulations, we use $T = 1/600$ s. The discretized dynamics are thus written as:

$$x(t + T) = Ax(t) + Bu(t),$$ (9)

where $A = I + A_c T$ and $B = B_c T$.

Finally, we perform a coordinate shift to error dynamics. This allows us to apply standard control techniques for trajectory tracking. We define the tracking error to be $y = q - q_d$, where $q_d$ is the desired state, and this error obeys the following dynamics:

$$y(t+1) = Ay(t) + Bu(t) + w(t) + w_{traj}(t) \tag{10a}$$

$$w_{traj}(t) = A(q_d(t) - \bar{q}) + \bar{q} - q_d(t+1) \tag{10b}$$

where $w$ is the external perturbation. $w_{traj}$ represents the effect of constantly changing trajectories — for example, if the desired trajectory at the current time-step is some value $a$, and the desired trajectory at the next time-step is some other value $b$, then this is equivalent to introducing a perturbation of $b - a$. Error $y$ is a column vector with length $n_y$ (8 for front legs, and 6 for the other legs), and input $u$ is a column vector with length $n_u = n_y/2$.

To include motor delay (known as *actuation* delay in controls literature) and sensory delay, we make use of augmented state formulations as introduced in **Stenberg et al., 2022**. Let the motor delay be $\mu$ steps. We re-define $u$ as the *intended* actuation, which is delayed by $\mu$ steps before it affects the state. We introduce a variable $a$ such that $a_i$ represents $u$ delayed by $i$ steps. $a$ is a column vector with length $\mu \times n_u$, and it can be written as:

$$a_1(t+1) = u(t), \quad a_i(t+1) = a_{i-1}(t), \quad i \in [2, \mu] \tag{11}$$

Similarly, let the sensory delay be $\delta$ steps. The state information $y$ is delayed by $\delta$ steps before it reaches the controller. We introduce another variable $s$ such that $s_i$ represents $y$ delayed by $i$ steps. $s$ is a column vector with length $\delta \times n_y$, and can be written as:

$$s_1(t+1) = y(t), \quad s_i(t+1) = s_{i-1}(t), \quad i \in [2, \delta] \tag{12}$$

We define $g$ as the effect of changing trajectories in the future, where $g_i = w_{traj}(t+i)$. Note that this does not correspond to any physical signal — rather, it is a virtual variable that allows us to incorporate knowledge of the future trajectory. $g$ is a column vector with length $(\mu + 1) \times n_y$, and it can be written as:

$$g_\mu(t+1) = w_{traj}(t+\mu+1), \quad g_i(t+1) = g_{i+1}(t), \quad i \in [0, \mu-1] \tag{13}$$

We can now rewrite the dynamics (**Equation 10a**) to include delays using these variables:

$$y(t+1) = Ay(t) + Ba_\mu(t) + g_0(t) + w(t) \tag{14}$$

Finally, we define $f$ as the prediction of $y$ in the future (assuming no perturbations), where $f_i$ represents the prediction $i$ steps into the future. Like $g$, this variable does not correspond to any physical signal, and is used to incorporate predictive capability into the controller. $f$ is a column vector of length $\mu \times n_y$. The first portion of $f$ can be written as

$$\begin{aligned}
f_1(t+1) &= Ay(t+1) + Ba_{\mu-1}(t) + g_1(t) \\
&= A^2 y(t) + ABa_\mu(t) + Ag_0(t) + Aw(t) + Ba_{\mu-1}(t) + g_1(t)
\end{aligned} \tag{15}$$

and subsequent values can be written as

$$f_i(t+1) = A(f_{i-1}(t+1)) + Ba_{\mu-i}(t) + g_i(t), \quad i \in [2, \mu-1] \tag{16}$$

$$f_\mu(t+1) = A(f_{\mu-1}(t+1)) + Bu(t) + g_\mu(t)$$

where the $A$ term must be written out and simplified as is done in **Equation 15**. We define an augmented state vector $z$:

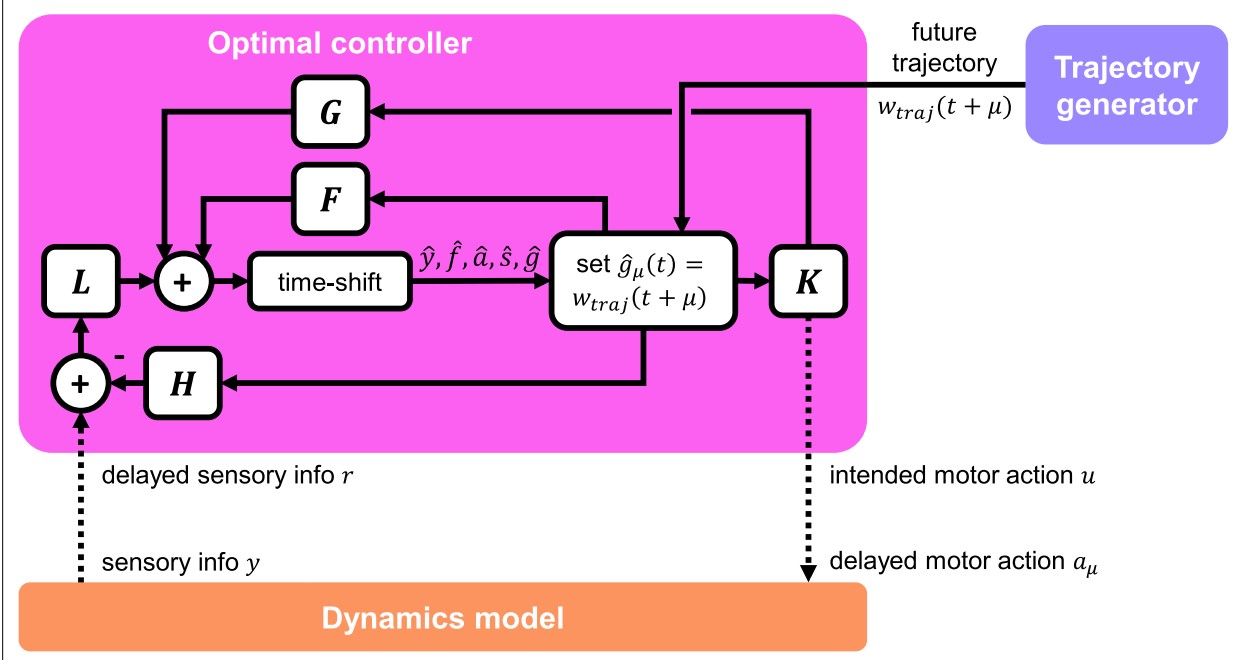

**Figure 6.** Detailed schematic of optimal controller. The controller receives delayed sensory information from the dynamics model, as well as future trajectory information from the trajectory generator. The controller interacts with the dynamics model via a delayed motor action signal. The internal structure of the controller is that of a standard output feedback controller, which incorporates estimation via a Kalman filter. Dotted lines indicate muscle and sensor delays.

$$z(t) = \begin{bmatrix} y(t) \\ f(t) \\ a(t) \\ s(t) \\ g(t) \end{bmatrix} \tag{17}$$

and write the overall system in the form of

$$z(t + 1) = Fz(t) + Gu(t) + w_{aug}(t), \tag{18}$$

$$r(t) = Hz(t), \tag{19}$$

where $F$, $G$, $H$, and $w_{aug}(t)$ can be directly obtained by rearranging *Equation 11*, *Equation 12*, *Equation 13*, *Equation 14*, *Equation 16*. In particular, $H$ is zero everywhere except at the block corresponding to $s_\delta$, where it is identity, i.e., $r(t) = s_\delta(t)$. This is the only information from the system that is received by the controller.

To achieve effective trajectory tracking, we seek a control law under which $y$ remains small. This can be achieved using the Linear Quadratic Gaussian (LQG) controller (*Åström and Murray, 2021*). The controller is governed by the following equations:

$$\hat{z}(t + 1) = F\hat{z}(t) + Gu(t) + L(r(t) - H\hat{z}(t)) \tag{20}$$

$$u(t) = K\hat{z}(t), \tag{21}$$

where $\hat{z}$ is the estimate of the augmented state (comprised of $\hat{y}, \hat{f}, \hat{a}, \hat{s}$, and $\hat{g}$), estimated via a steady-state Kalman filter; $L$ and $K$ are the optimal observer and controller matrices, respectively, synthesized via discrete algebraic Riccati equations. We directly feed in future values of the trajectory by setting $\hat{g}_\mu(t) = w_{traj}(t + \mu)$. By doing this, we ensure that the future trajectory 'estimate' is perfect, i.e., $\hat{g} = g$. This helps the controller estimate values of $f$ (future states) and $y$. The overall information flow within, to, and from the controller is shown in *Figure 6*.

### Generate joint trajectories of the complete model with perturbations

The full model integrates all of the modules to generate trajectories of joint angles over time. The phase coordinator and trajectory generators combine to compute the desired joint angles $\theta_i$ for each leg $i$, and the controller implements them constrained by the dynamics. We run the full model at 600 Hz.

At each time step, we first update the phases of the legs $\phi_i$ based on the phase coupling equation above. Every 2 timesteps, we update the target joint angles $\theta_d$ using the trajectory generator model. We run the trajectory generator $F$ for $d_{\mathrm{motor}}$ time steps on its own, by continually integrating its output. This future trajectory forms the basis of $f$, the predicted future errors used to guide the controller.

Each time step, we run the controller and dynamics model to control the torque $\tau$ so that the joint angles $\theta$ go towards the target joint angles $\theta_d$. If there is a disturbance, we apply it to the joint angles and derivatives at this point. Every 8 timesteps, we set $\theta_d = \theta$, so that the trajectory generator predicts an intended trajectory in line with the current state.

For the perturbation numerical experiments described in the Results, we ran all simulations for 1800 timesteps at 600 Hz. For persistent stochastic perturbations, we applied the perturbation from $600^{\mathrm{th}}$ timesteps to $1200^{\mathrm{th}}$ timesteps. For impulse perturbations, we applied a single strong perturbation at $600^{\mathrm{th}}$ timesteps.

We ran all our simulations on an Intel(R) Core(TM) i9-9940X CPU. We used GNU Parallel (*Tange, 2011*) to run simulations on multiple cores simultaneously.

### Computing kinematic similarity (KS) by quantifying likelihood of walking kinematics relative to ground truth

We followed a multi-step procedure in order to quantify the likelihood of the simulated walking kinematics relative to observed distribution of walking kinematics. This procedure is schematized in *Figure 4A*.

To fit our likelihood model, we first performed Principal Components Analysis (PCA) to reduce the tracked joint angles from our 3D kinematics data to two dimensions per frame. To account for the circular nature of rotation angles, we performed PCA on the combination of sines and cosines of each angle. Next, we used a Gaussian Kernel Density Estimation (KDE) to estimate the probability density function of the principal components. Thus, we obtained a likelihood model for joint angle kinematics at each frame.

We chose two dimensions for PCA for two key reasons. First, these two dimensions alone accounted for a large portion of the variance in the data (52.7% total, with 42.1% for the first component and 10.6% for the second component). There was a big drop in variance explained from the first to the second component, but no sudden drop in the next 10 components (see Appendix 8). Second, the KDE procedure only works effectively in low-dimensional spaces, and the minimal number of dimensions needed to obtain circular dynamics for walking is 2. We investigate the effect of varying the number of dimensions of PCA in Appendix 9.

We run our model described above to produce simulated joint angle trajectories. To estimate a likelihood of a simulated set of angles, we first projected them onto the same principal components identified from the observed kinematics. Then, we use the KDE model to estimate the logarithmic probability density function (log PDF) for each frame during the perturbation; we refer to this as the *kinematic similarity (KS)*. For persistent stochastic perturbations, we estimated mean KS during the perturbation, from $600^{\mathrm{th}}$ timesteps to $1200^{\mathrm{th}}$ timesteps. For impulse perturbations, we estimated mean KS in the transient recovery process, which we estimated to be from $610^{\mathrm{th}}$ timesteps to the $800^{\mathrm{th}}$ timesteps.

### Visualization of joint movement trajectories

For *Video 1*, we visualized the simulated and real joint movements using the biomechanical fly body model from *Vaxenburg et al., 2024*. For each frame, we ran an inverse kinematics optimization over the model angles to match the simulated or real joint positions of the fly. We did not simulate realistic physics of the fly legs and their interactions with the ground in these visualizations. In our data, the fly thorax was fixed, and the wings were removed, so in these visualizations, we also fixed every other degree-of-freedom in the fly model besides the six legs. For the remaining videos, we visualized the joints as ball-and-stick models using matplotlib (*Hunter, 2007*).

## Acknowledgements

We thank J Doyle for initial discussions on layered feedback models, N Csomay-Shanklin for initial discussions on deriving models of legged locomotion, A Nair for discussions around coupled oscillator models, A Schwarze for discussions around sensorimotor feedback loops, E Dickinson for technical assistance, and members of the Tuthill and Brunton labs for helpful discussions and feedback on this manuscript. We are grateful to R Vaxenburg, I Siwanowicz, and S Turaga for help adapting their virtual fly model for visualizing real and simulated fly walking. This work has been funded by a National Science Foundation Graduate Research Fellowship to LK; a NSERC Postgraduate Scholarship (NSERC PGSD3-557385-2021) to JSL; a Searle Scholar Award, a Klingenstein-Simons Fellowship, a Pew Biomedical Scholar Award, a McKnight Scholar Award, a Sloan Research Fellowship, the New York Stem Cell Foundation, and NIH R01NS102333 to JCT; JCT is a New York Stem Cell Foundation – Robertson Investigator; Air Force Office of Scientific Research award FA9550-19-1-0386 and the Richard & Joan Komen University Chair to BWB.

## Additional information

### Competing interests

John C Tuthill: Reviewing editor, eLife. The other authors declare that no competing interests exist.

### Funding

| Funder | Grant reference number | Author |
| --- | --- | --- |
| National Science Foundation | Graduate Research Fellowship Program | Lili Karashchuk |
| Natural Sciences and Engineering Research Council of Canada | PGSD3-557385-2021 | Jing Shuang Li |
| Searle Scholars Program | | John C Tuthill |
| Klingenstein-Simons Fellowship | | John C Tuthill |
| Pew Charitable Trusts | | John C Tuthill |
| McKnight Endowment Fund for Neuroscience | | John C Tuthill |
| Alfred P. Sloan Foundation | Sloan Research Fellowship | John C Tuthill |
| New York Stem Cell Foundation | | John C Tuthill |
| National Institutes of Health | R01NS102333 | John C Tuthill |
| Air Force Office of Scientific Research | FA9550-19-1-0386 | Steven L Brunton Bingni W Brunton |
| Richard & Joan Komen University Chair | | Bingni W Brunton |

The funders had no role in study design, data collection and interpretation, or the decision to submit the work for publication.

### Author contributions

Lili Karashchuk, Conceptualization, Data curation, Software, Formal analysis, Visualization, Writing – original draft, Writing – review and editing; Jing Shuang Li, Conceptualization, Software, Formal analysis, Writing – original draft, Writing – review and editing; Grant M Chou, Data curation, Visualization, Methodology, Writing – review and editing; Sarah Walling-Bell, Methodology; Steven L Brunton, Conceptualization, Funding acquisition, Writing – review and editing; John C Tuthill, Bingni W Brunton,

Conceptualization, Formal analysis, Supervision, Funding acquisition, Investigation, Writing – original draft, Project administration, Writing – review and editing

**Author ORCIDs**
Lili Karashchuk ⓘ https://orcid.org/0000-0001-6244-8239
Jing Shuang Li ⓘ https://orcid.org/0000-0003-4931-8709
John C Tuthill ⓘ https://orcid.org/0000-0002-5689-5806
Bingni W Brunton ⓘ https://orcid.org/0000-0002-4831-3466

Reviewer #1 (Public Review): https://doi.org/10.7554/eLife.99005.3.sa1
Reviewer #2 (Public Review): https://doi.org/10.7554/eLife.99005.3.sa2
Author response https://doi.org/10.7554/eLife.99005.3.sa3

## Additional files

### Supplementary files
MDAR checklist

### Data availability
The current manuscript is a computational study, so no data have been generated for this manuscript. All code is openly available https://github.com/lambdaloop/layered-walking/ (copy archived at *Li and Karashchuk, 2025*).

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

## Appendix 1

## Velocity and phase distributions in data

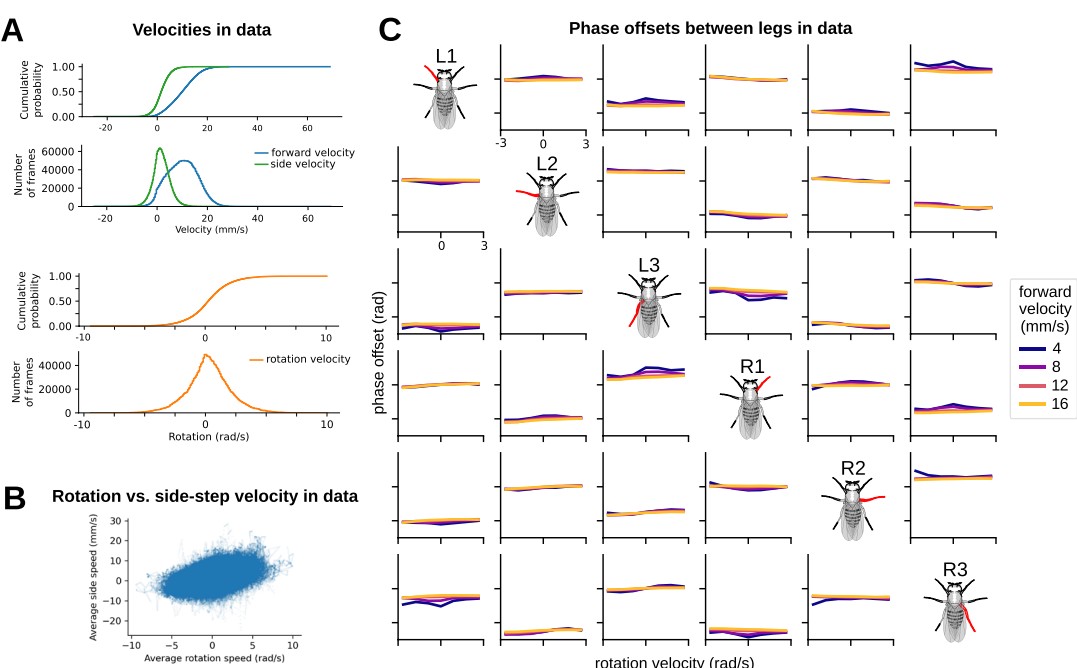

**Appendix 1—figure 1.** Velocity and phase distributions in data justify the selection of model simulation parameters. (**A**) Cumulative distribution functions and number of frames associated with various forward, rotation, and side-step velocities in data. Right/left (side-step) walking peaked at 0 mm/s, while forward/backward walking achieved a sustained peak from approximately 4 mm/s to 10 mm/s. Rotation velocity peaked at 0 rad/s. Most model-generated simulations in the paper adhered to this range. (**B**) Rotation vs. side-step velocity. Each dot on the scatter plot represents a single bout from data. Rotation velocity and side-step velocity were highly correlated with one another in data. (**C**) Phase offsets between legs in data over a range of walking velocities and rotational velocities. Side-step velocities were omitted from the study since (as previously demonstrated) they were highly correlated with rotation velocity. Phase offsets remained relatively constant over a range of forward and rotational velocities.

## Appendix 2

## Angle vs. phase plots for all legs and joints

These plots are supplementary to *Figure 2B*.

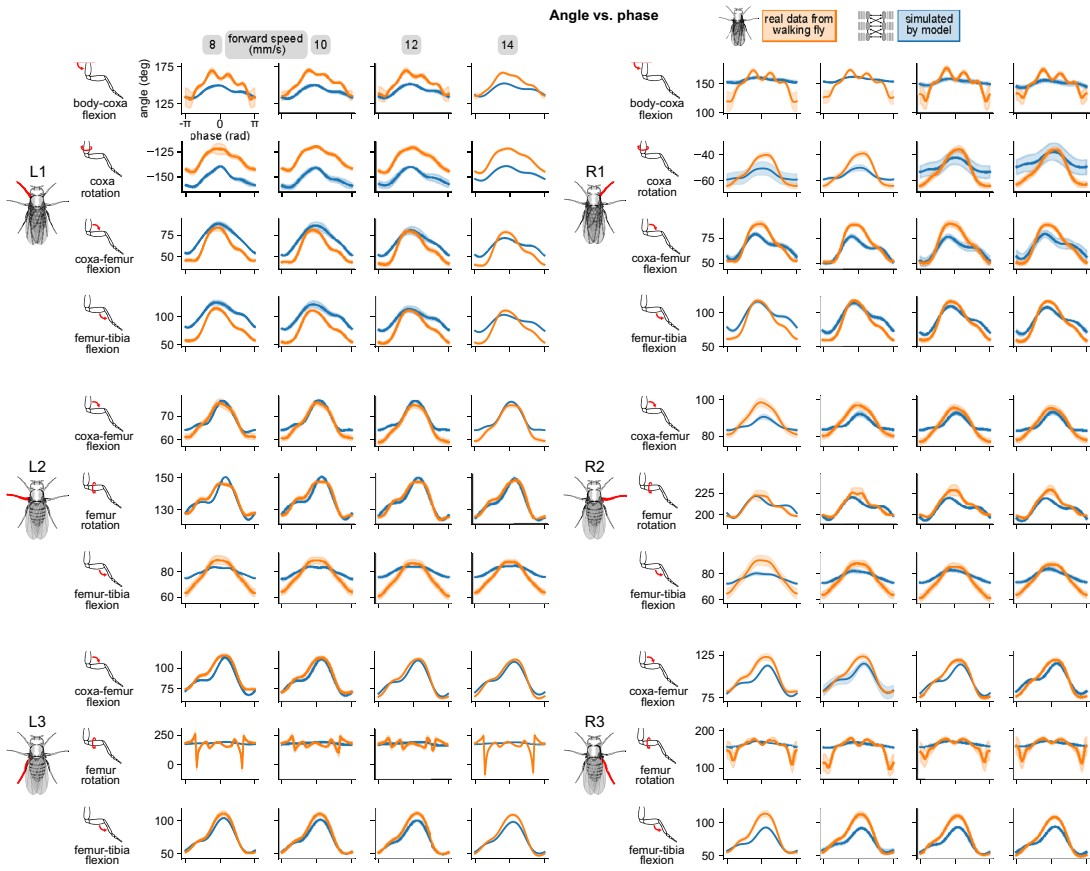

**Appendix 2—figure 1.** Simulated walking reproduced realistic joint angles. Angle vs. computed per-leg phase of all joints on all legs for four different walking speeds. All simulations used a sensory delay of 10 ms and motor delay of 30 ms, consistent with experimental values. Some oddities were observed in real data, for instance in L3 femur flexion and R1 body-coxa flexion. The model does not quite capture these oddities.

## Appendix 3

### Angular velocity vs. phase plots for all legs and joints

These plots are supplementary to *Figure 2D*.

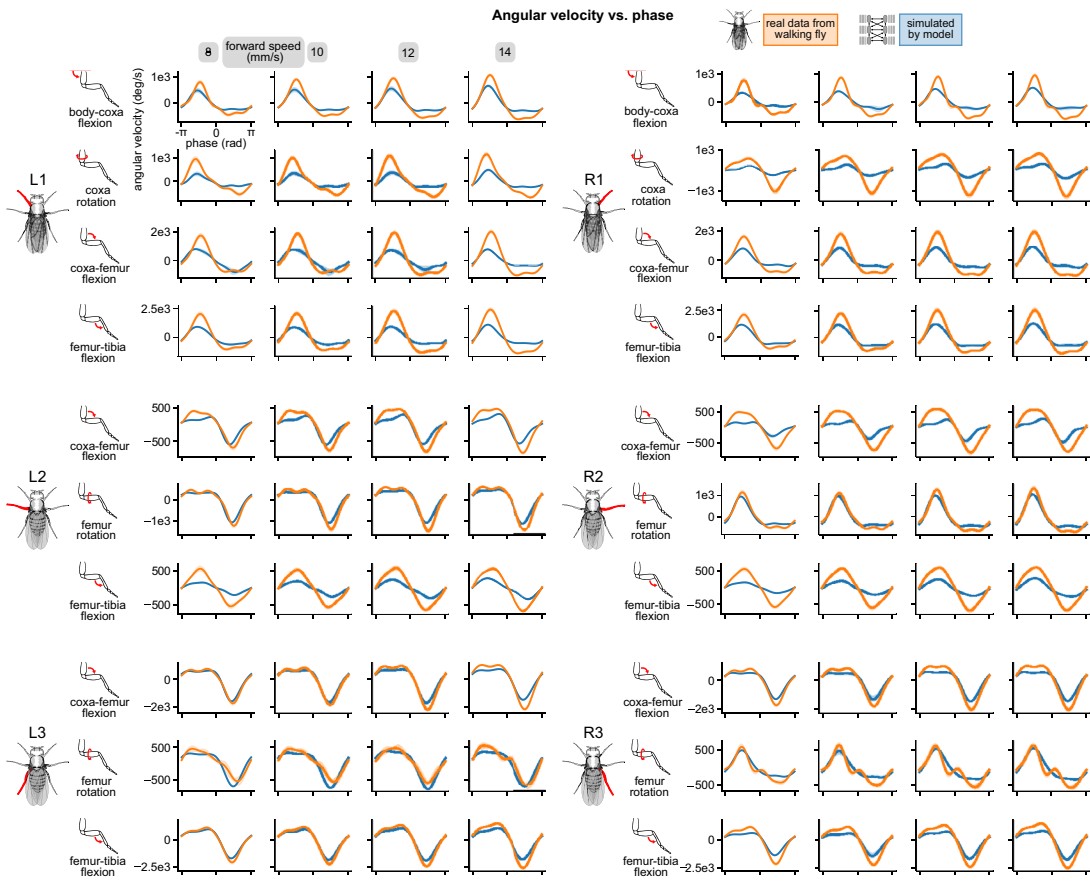

**Appendix 3—figure 1.** Simulated walking reproduced realistic joint angular velocities. Angular velocity vs. computed per-leg phase of all joints on all legs for four different walking speeds. All simulations used a sensory delay of 10 ms and motor delay of 30 ms, consistent with experimental values.

## Appendix 4

### Differences between model and data

These plots are supplementary to *Figure 2E*.

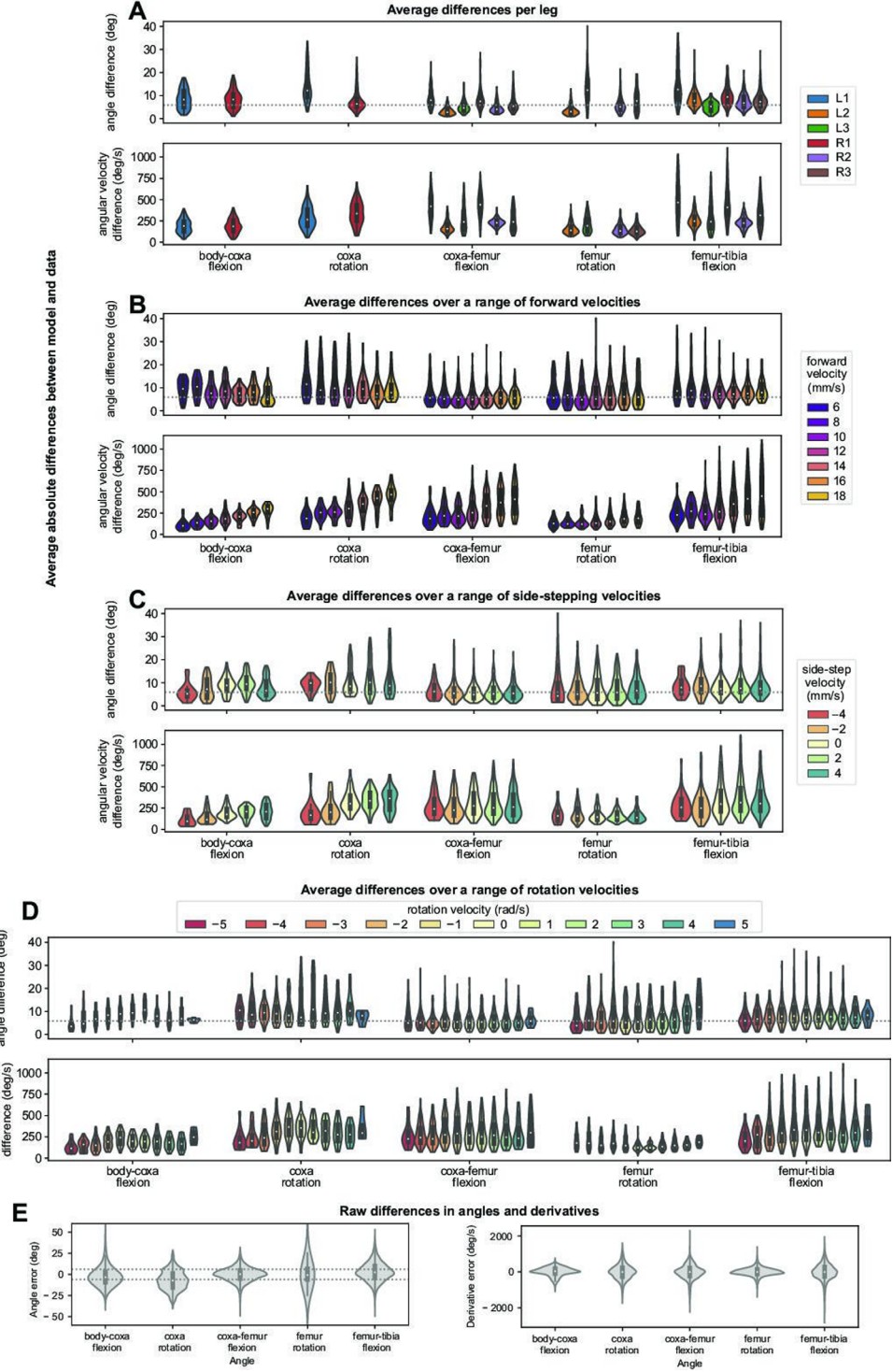

**Appendix 4—figure 1.** Simulated walking resembled data over a range of forward, turning, and side-stepping velocities. Average differences between model simulations over 500 distinct bouts. All simulations use a sensory delay of 10 ms and motor delay of 30 ms, consistent with experimental values. The dotted line (5.56 degrees) indicates uncertainty associated with data collection.

## Appendix 5

## Phase coupling within and across legs

We quantify the model's capacity to simulate leg kinematics by comparing the phase coupling of simulated joint kinematics with real walking flies, both within and across legs. A phase difference probability density peak close to zero between two joint angles means that they are strongly coupled, whereas a broader peak means they are weakly coupled.

To assess the effectiveness of the model's inter-leg coordination, we compared phase coupling across legs for simulated vs. real walking. We found that the model reproduced patterns in the data, with some exceptions. Here, a phase difference of zeros means the two legs are synchronized, whereas a phase difference of $\pi$ means the two legs move in anti-phase. Fly data showed largely tripod coordination, so that L1-R2-L3 are synchronized with each other and also anti-phase with R1-L2-R3. The model showed coupling properties that were qualitatively similar to the coupling properties of the data, with moderate variations in phase difference (i.e. peak location) and synchronization strength (i.e. peak height). Some minor differences may also be due to violations of our assumption of constant phase offset across forward walking speeds. This assumption holds true over a large range of speeds, but breaks down at the lowest walking speeds (see Appendix 1). In the model, the phase coordinator is responsible for representing inter-leg coupling; these results suggest that the underlying Kuramoto oscillator reproduces naturalistic coupling patterns.

Next, we examined within-leg coupling in data, where phase synchronizations between different joints on the same leg can be both strong (e.g. L3 coxa-femur flexion to femur-tibia flexion) and weak (e.g. R2 femur-tibia flexion to femur rotation). Despite the mirror symmetry of the fly, within-leg coupling is not necessarily symmetric in the data — for instance, compare L2 and R2 coxa-femur flexion. All within-leg synchronizations (i.e. peaks) in data were reproduced by the model; however, the model generally exhibited stronger synchronization (i.e. higher peaks) than the data. In some cases (e.g. L3 femur rotation to femur-tibia flexion), the model exhibited synchronizations that are not present in the data. Overall, the model generally exhibits stronger within-leg coupling when compared to the data.

The trajectory generator is responsible for reproducing the per-leg inter-joint coupling, so these results suggest that the neural network learned stronger coupling values than are present in the data. This result is consistent with the time series comparisons, where we observed that the model produced more regular joint angle trajectories than the fly (*Figure 2A*).

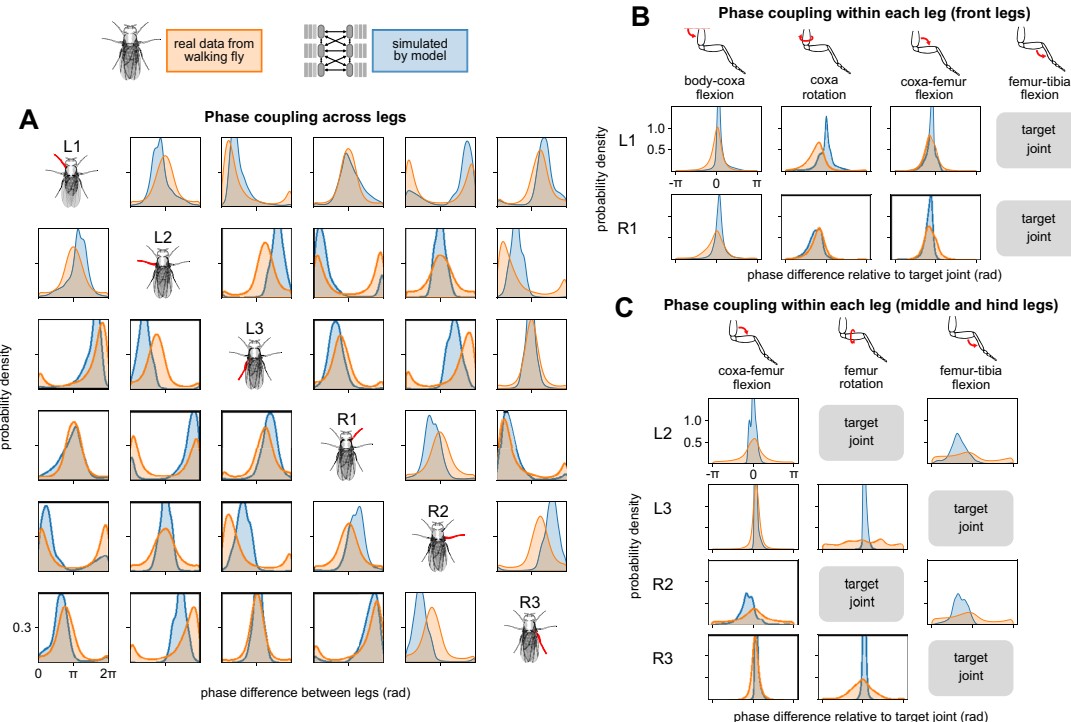

**Appendix 5—figure 1.** Simulated walking reproduced across- and within-leg coupling characteristics of data. (**A**) Phase coupling across legs. We compared phases of representative joints across legs. Model coupling qualitatively resembled data coupling, with moderate variations in phase difference (i.e. peak location) and synchronization strength (i.e. peak height). A strong peak in probability density indicates synchronization, whereas broader peaks indicate weak synchronization. A single peak at zero on the horizontal axis indicates that the two joint phases are coupled to match; a single peak elsewhere indicates that the two joint phases are coupled with some phase offset. Simulations were performed over a range of forward walking, turning, and side-stepping speeds over 500 distinct bouts. (**B, C**) Phase coupling within each leg. For each leg, we compared phases between a representative joint for the leg (denoted 'target' on the image) and other joints on the leg. Gaps are present as we did not include all five joints for all legs in the model. All synchronizations (i.e. peaks) in data are reproduced by the model. However, simulations generally exhibited stronger synchronization (i.e. higher peaks).

## Appendix 6

### Estimate of plausible stochastic perturbation values

Here, we derive an estimate of the maximum plausible stochastic perturbation magnitude corresponding to a sudden gust of wind. Wind force experienced by a given leg is

$$F = \frac{1}{2}\rho v^2 A, \tag{22}$$

where $\rho$, $v$, and $A$ are the density of air, wind speed, and perpendicular area (to wind direction) of the leg. The resulting acceleration is

$$a = \frac{F}{m} = \frac{1}{2m}\rho v^2 A, \tag{23}$$

where $m$ is the mass of the fly.

We assume, generously, that the leg experiences the same acceleration. Since the controller operates at a frequency of $f$, we can approximate the angular velocity (i.e. perturbation) $\omega$ generated by the slip acceleration in one timestep for one joint on the leg as

$$\omega = \frac{a}{fr} = \frac{1}{2mfr}\rho v^2 A, \tag{24}$$

where $r$ is the length of the leg.

Substituting in values used in our model ($m$=0.7e-6 kg, $f$ = 600 Hz, r = 0.0015 m), and let $\rho$ = 1.2 kg/m$^3$ (standard value), $v$ = 6 m/s (corresponding to a strong 21 km/hr wind), and $A$ = 0.15e-6 m$^2$ (corresponding to a 1.5 mm by 0.1 mm leg), we obtain a maximum joint perturbation magnitude of $\omega$ = 5.1 rad/s.

### Estimate of plausible impulse perturbation values

We estimate the maximum plausible impulse perturbation corresponding to a sudden slip. Assume that a leg steps at an angle of $\theta$ (from the vertical) onto a flat, frictionless ground. The resulting horizontal acceleration would be

$$a = g\tan\theta, \tag{25}$$

where $g$ is acceleration due to gravity.

Assume the controller operates at a frequency of $f$. We can approximate the angular velocity (i.e. perturbation) $\omega$ generated by the slip acceleration in one timestep for a joint on the leg as

$$\omega = \frac{a}{fr} = \frac{g\tan\theta}{fr}, \tag{26}$$

where $r$ is the length of the leg. Plugging in values of $g$ = 9.81 m/s$^2$ (standard value), $f$ = 600 Hz (value used in the model), $r$ = 0.0015 m (value used in the model), and $\theta$ = 45 degrees (quite a large slip angle), we obtain a perturbation value of n $\omega$ = 10.9 rad/s.

## Appendix 7

## Effect of sensory and motor delays on walking under impulse perturbations

The results shown in *Figure 5* correspond to persistent stochastic perturbations. Here, we present similar results for impulse perturbations here. Maximum values of motor and sensory delay values remain reasonably close to known values in physiology.

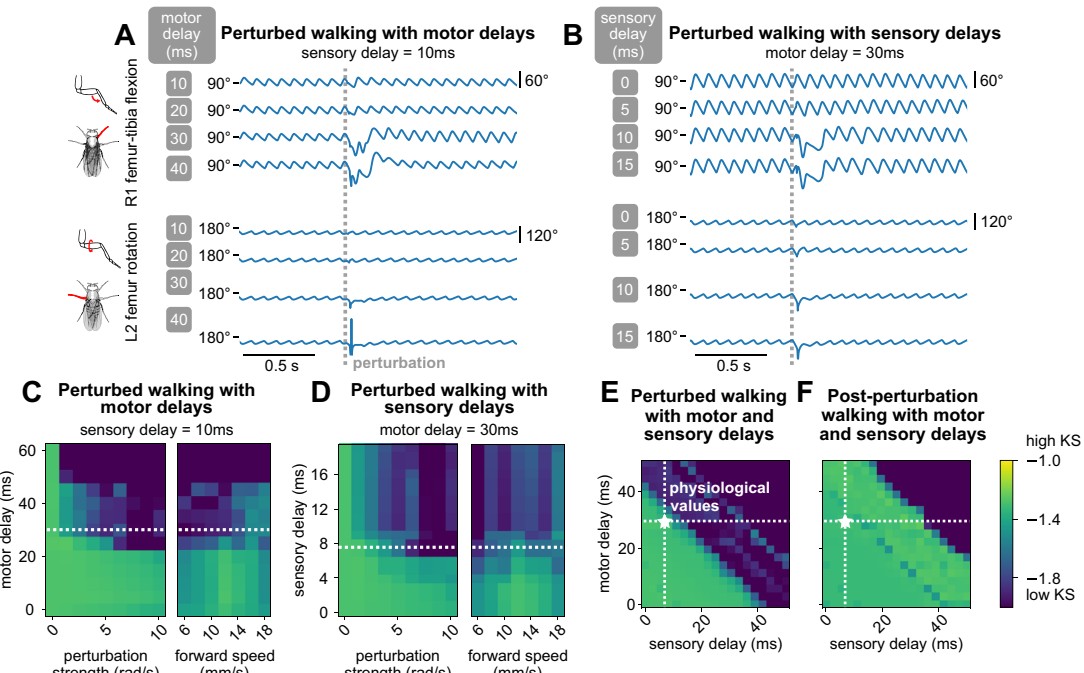

**Appendix 7—figure 1.** Model generated robust walking under persistent stochastic perturbations over select ranges of motor and sensory delays, revealing fundamental constraints on delay values. (**A, B**) Example simulated time series of femur-tibia flexion on leg R1 and femur rotation on leg L2 under various values of motor (10, 20, 30, 40 ms) and sensory delay (0, 5, 10, 15 ms). Perturbation effects were more noticeable with increased delay values. (**C, D, E**) KS of simulated during-perturbation walking to data, for various values of delay, perturbation strength, and forward speeds. For each square of the heatmap, four simulations with different initial conditions were simulated and evaluated. As perturbation strength and delays increased, simulated walking became less similar to data; the effect was more pronounced with increased delays. When we fixed one delay value and varied the other, the model maintained realistic walking (KS > −1.6) up to about 25 ms of motor delay and 8 ms of sensory delay across a range of conditions. When we allowed both motor and sensory delay to vary, the model maintained realistic walking when the sum of the delays was no more than about 40 ms. (**F**) Post-perturbation walking with motor and sensory delays. The model is able to recover from perturbations for very large values of delay. Unless otherwise stated, forward speed = 12 mm/s, and perturbation strength = 5 rad/s.

## Appendix 8

# Characterization of principal components of kinematics

These plots are supplementary to *Figure 4*.

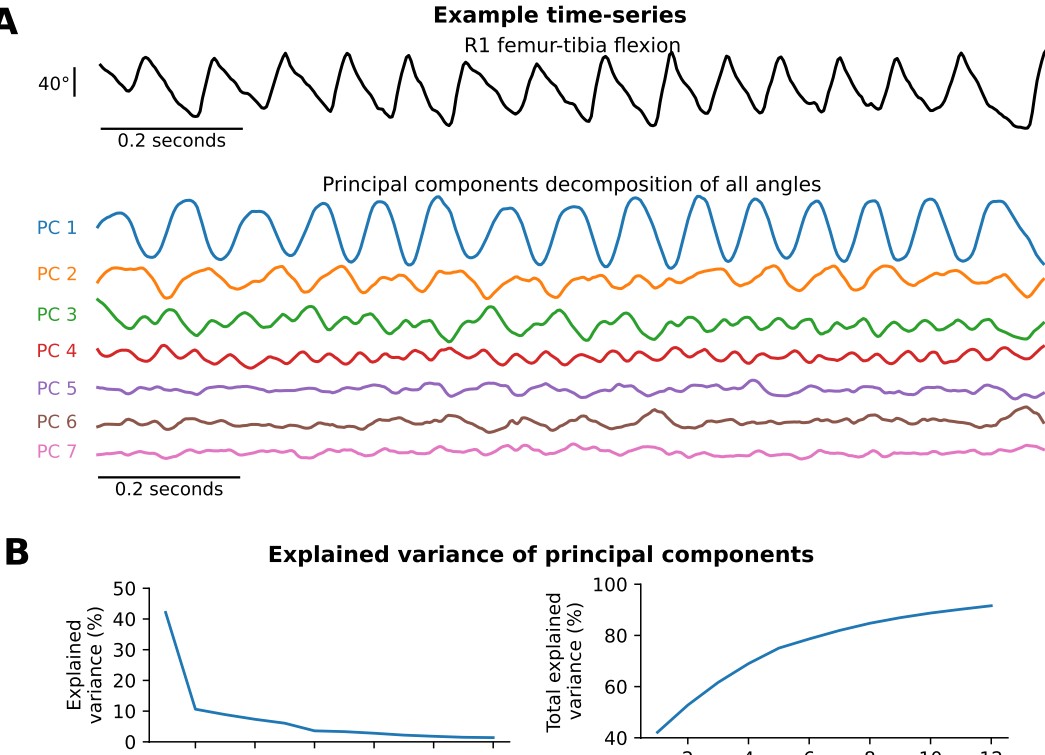

**Appendix 8—figure 1.** Principal components model the range of kinematics observed in the data. (**A**) An example time series of R1 femur-tibia flexion along with principal components of all the angles. The first two principal components (PC1 and PC2) capture the step cycle of walking. The later principal components capture increasingly higher-frequency aspects of the angle trajectories. (**B**) Explained variance of the principal components. Seven components explain over 81% of the variance and 12 components explain over 91% of the variance in joint angles.

## Appendix 9

## Effect of PCA dimension in kinematic similarity metric

These plots are supplementary to *Figure 5*.

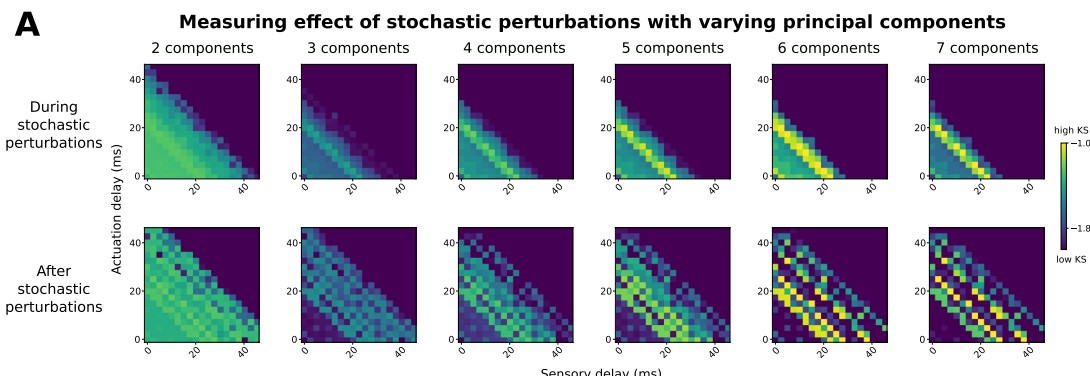

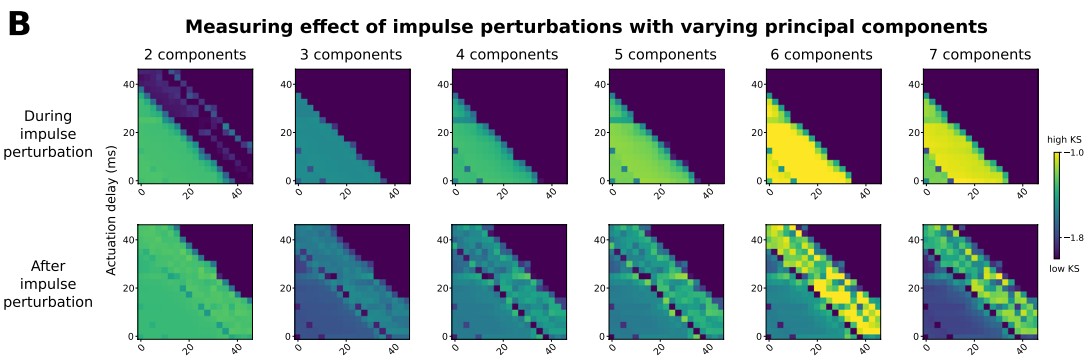

**Appendix 9—figure 1.** Results on sensory delay thresholds are generally robust to varying principal components analysis (PCA) dimension used in the kinematic similarity metric. Effect of varying principal component number on kinematic similarity (KS) measurement for stochastic (**A**) and impulse (**B**) perturbations. The results look similar across components for impulse perturbations. For stochastic perturbations, the range of similar walking decreases as we increase the number of components used to evaluate walking kinematics. As higher principal components represent higher temporal frequencies, it is likely that higher frequencies are impacted at the edge of stability before walking collapses entirely at increasing delays.

