## [Editor Report · eLife Assessment]

This **valuable** study presents a computational model that simulates walking motions in *Drosophila* and suggests that, if sensorimotor delays in the neural circuitry were any longer, the system would be easily destabilized by external perturbations. The hierarchical control model is sensible and the evidence supporting the conclusions **convincing**. The modular model, which has many interacting components with varying degrees of biological realism, will serve as a well-grounded starting point for future studies that incorporate richer or more complete empirical data.

---

## [Referee Report · Reviewer #1 (Public Review)]

Summary:

In this work, the authors present a novel, multi-layer computational model of motor control to produce realistic walking behaviour of a *Drosophila* model in the presence of external perturbations and under sensory and motor delays. The novelty of their model of motor control is that it is modular, with divisions inspired by the fly nervous system, with one component based on deep learning while the rest are based on control theory. They show that their model can produce realistic walking trajectories. Given the mostly reasonable assumptions of their model, they convincingly show that the sensory and motor delays present in the fly nervous system are the maximum allowable for robustness to unexpected perturbations.

Their fly model outputs torque at each joint in the leg, and their dynamics model translates these into movements, resulting in time-series trajectories of joint angles. Inspired by the anatomy of the fly nervous system, their fly model is a modular architecture that separates motor control at three levels of abstraction:

(1) oscillator-based model of coupling of phase angles between legs,

(2) generation of future joint-angle trajectories based on the current state and inputs for each leg (the trajectory generator), and

(3) closed-loop control of the joint-angles using torques applied at every joint in the model (control and dynamics).

These three levels of abstraction ensure coordination between the legs, future predictions of desired joint angles, and corrections to deviations from desired joint-angle trajectories. The parameters of the model are tuned in the absence of external perturbations using experimental data of joint angles of a tethered fly. A notable disconnect from reality is that the dynamics model used does not model the movement of the body and ground contacts as is the case in natural walking, nor the movement of a ball for a tethered fly, but instead something like legs moving in the air for a tethered fly.

In order to validate the realism of the generated simulated walking trajectories, the authors compare various attributes of simulated to real tethered fly trajectories and show qualitative and quantitative similarities, including using a novel metric coined as Kinematic Similarity (KS). The KS score of a trajectory is a measure of the likelihood that the trajectory belongs to the distribution of real trajectories estimated from the experimental data. While such a metric is a useful tool to validate the quality of simulated data, there is some room for improvement in the actual computation of this score. For instance, the KS score is computed for any given time-window of walking simulation using a fraction of information from the joint-angle trajectories. It is unclear if the remaining information in joint-angle trajectories that are not used in the computation of the KS score can be ignored in the context of validating the realism of simulated walking trajectories.

The authors validate simulated walking trajectories generated by the trained model under a range of sensorimotor delays and external perturbations. The trained model is shown to generate realistic joint-angle trajectories in the presence of external perturbations as long as the sensorimotor delays are constrained within a certain range. This range of sensorimotor delays is shown to be comparable to experimental measurements of sensorimotor delays, leading to the conclusion that the fly nervous system is just fast enough to be robust to perturbations.

Strengths:

This work presents a novel framework to simulate *Drosophila* walking in the presence of external perturbations and sensorimotor delay. Although the model makes some simplifying assumptions, it has sufficient complexity to generate new, testable hypotheses regarding motor control in *Drosophila*. The authors provide evidence for realistic simulated walking trajectories by comparing simulated trajectories generated by their trained model with experimental data using a novel metric proposed by the authors. The model proposes a crucial role in future predictions to ensure robust walking trajectories against external perturbations and motor delay. Realistic simulations under a range of prediction intervals, perturbations, and motor delays generating realistic walking trajectories support this claim. The modular architecture of the framework provides opportunities to make testable predictions regarding motor control in *Drosophila*. The work can be of interest to the *Drosophila* community interested in digitally simulating realistic models of *Drosophila* locomotion behaviors, as well as to experimentalists in generating testable hypotheses for novel discoveries regarding neural control of locomotion in *Drosophila*. Moreover, the work can be of broad interest to neuroethologists, serving as a benchmark in modelling animal locomotion in general.

Weaknesses:

As the authors acknowledge in their work, the control and dynamics model makes some simplifying assumptions about *Drosophila* physics/physiology in the context of walking. For instance, the model does not incorporate ground contact forces and inertial effects of the fly's body. It is not clear how these simplifying assumptions would affect some of the quantitative results derived by the authors. The range of tolerable values of sensorimotor delays that generate realistic walking trajectories is shown to be comparable with sensorimotor delays inferred from physiological measurements. It is unclear if this comparison is meaningful in the context of the model's simplifying assumptions. The authors propose a novel metric coined as Kinematic Similarity (KS) to distinguish realistic walking trajectories from unrealistic walking trajectories. Defining such an objective metric to evaluate the model's predictions is a useful exercise, and could potentially be applied to benchmark other computational animal models that are proposed in the future. However, the KS score proposed in this work is calculated using only the first two PCA modes that cumulatively account for less than 50% of the variance in the joint angles. It is not obvious that the information in the remaining PCA modes may not change the log-likelihood that occurs in the real walking data.

Comments on revisions:

The authors have addressed the concerns and questions raised in the original review.

---

## [Referee Report · Reviewer #2 (Public Review)]

Summary:

In this study, Karashchuk et al. develop a hierarchical control system to control the legs of a dynamic model of the fly. They intend to demonstrate that temporal delays in sensorimotor processing can destabilize walking and that the fly's nervous system may be operating with as long of delays as could possibly be corrected for.

Strengths:

Overall, the approach the authors take is impressive. Their model is trained using a huge dataset of animal data, which is a strength. Their model was not trained to reproduce animal responses to perturbations, but it successfully rejects small perturbations and continues to operate stably. Their results are consistent with the literature, that sensorimotor delays destabilize movements.

Weaknesses:

The model is sophisticated and interesting, but the reviewer has great concerns regarding this manuscript's contributions, as laid out in the abstract:

(1) Much simpler models can be used to show that delays in sensorimotor systems destabilize behavior (e.g., Bingham, Choi, and Ting 2011; Ashtiani, Sarvestani, and Badri-Sproewitz 2021), so why create this extremely complex system to test this idea? The complexity of the system obscures the results and leaves the reviewer wondering if the instability is due to the many, many moving parts within the model. The reviewer understands (and appreciates) that the authors tested the impact of the delay in a controlled way, which supports their conclusion. However, the reviewer thinks the authors did not use the most parsimonious model possible, and as such, leave many possible sources for other causes of instability.

(2) In a related way, the reviewer is not sure that the elements the authors introduced reflect the structure or function of the fly's nervous system. For example, optimal control is an active field of research and is behind the success of many-legged robots, but the reviewer is not sure what evidence exists that suggests the fly ventral nerve cord functions as an optimal controller. If this were bolstered with additional references, the reviewer would be less concerned.

(3) "The model generates realistic simulated walking that matches real fly walking kinematics...". The reviewer appreciates the difficulty in conducting this type of work, but the reviewer cannot conclude that the kinematics "match real fly walking kinematics". The range of motion of several joints is 30% too small compared to the animal (Figure 2B) and the reviewer finds the video comparisons unpersuasive. The reviewer would understand if there were additional constraints, e.g., the authors had designed a robot that physically could not complete the prescribed motions. However the reviewer cannot think of a reason why this simulation could not replicate the animal kinematics with arbitrary precision, if that is the goal.

Comments on revisions:

The authors have addressed the concerns and questions raised in the original review.

---

## [Author Response]

The following is the authors’ response to the original reviews.

We thank the editor and reviewers for their supportive comments about our modeling approach and conclusions, and for raising several valid concerns; we address them briefly below. In addition, a detailed, point-by-point response to the reviewers’ comments are below, along with additions and edits we have made to the revised manuscript.

Concerns about model’s biological realism and impact on interpretations

The goal of this paper was to use an interpretable and modular model to investigate the impact of varying sensorimotor delays. Aspects of the model (e.g. layered architecture, modularity) are inspired by biology; at the same time, necessary abstractions and simplifications (e.g. using an optimal controller) are made for interpretability and generalizability, and they reflect common approaches from past work. The hypothesized effects of certain simplifying assumptions are discussed in detail in Section 3.5. Furthermore, the modularity of our model allows us to readily incorporate additional biological realism (e.g. biomechanics, connectomics, and neural dynamics) in future work. In the revision, we have added citations and edits to the text to clarify these points.

Concerns that the model is overly complex

To investigate the impact of sensorimotor delays on locomotion, we built a closed-loop model that recapitulates the complex joint trajectories of fly walking. We agree that locomotion models face a tradeoff between simplicity/interpretability and realism — therefore, we developed a model that was as simple and interpretable as possible, while still reasonably recapitulating joint trajectories and generalizing to novel simulation scenarios. Along these lines, we also did not select a model that primarily recreates empirical data, as this would hinder generalizability and add unnecessary complexity to the model. We do not think these design choices are significant weaknesses of this model; in fact, few comparable models account for all joints involved in locomotion, and fewer explicitly compare model kinematics with kinematics from data. We have add citations and edits to the text to clarify these points in the revision.

Concerns about the validity of the Kinematic Similarity (KS) metric to evaluate walking

We chose to incorporate only the first two PCA modes dimensions in the KS metric because the kernel density estimator performs poorly for high dimensional data. Our primary use of this metric was to indicate whether the simulated fly continues walking in the presence of perturbations. For technical reasons, it is not feasible to perform equivalent experiments on real walking flies, which is one of the reasons we explore this phenomenon with the model. We note the dramatic shift from walking to nonwalking as delay increases (Figure 5). To be thorough, in the revision, we have investigated the effect of incorporating additional PCA modes, and whether this affects the interpretation of our results. We have additionally added to the discussion and presentation of the KS metric to clarify its purpose in this study. We agree with the reviewers that the KS metric is too coarse to reflect fine details of joint kinematics; indeed, in the unperturbed case, we evaluate our model’s performance using other metrics based on comparisons with empirical data (Figures 2, 7, 8).

**Public Reviews:**

**Reviewer #1 (Public Review):**
Summary:In this work, the authors present a novel, multi-layer computational model of motor control to produce realistic walking behaviour of a *Drosophila* model in the presence of external perturbations and under sensory and motor delays. The novelty of their model of motor control is that it is modular, with divisions inspired by the fly nervous system, with one component based on deep learning while the rest are based on control theory. They show that their model can produce realistic walking trajectories. Given the mostly reasonable assumptions of their model, they convincingly show that the sensory and motor delays present in the fly nervous system are the maximum allowable for robustness to unexpected perturbations.Their fly model outputs torque at each joint in the leg, and their dynamics model translates these into movements, resulting in time-series trajectories of joint angles. Inspired by the anatomy of the fly nervous system, their fly model is a modular architecture that separates motor control at three levels of abstraction:(1) oscillator-based model of coupling of phase angles between legs,(2) generation of future joint-angle trajectories based on the current state and inputs for each leg (the trajectory generator), and(3) closed-loop control of the joint-angles using torques applied at every joint in the model (control and dynamics).These three levels of abstraction ensure coordination between the legs, future predictions of desired joint angles, and corrections to deviations from desired joint-angle trajectories. The parameters of the model are tuned in the absence of external perturbations using experimental data of joint angles of a tethered fly. A notable disconnect from reality is that the dynamics model used does not model the movement of the body and ground contacts as is the case in natural walking, nor the movement of a ball for a tethered fly, but instead something like legs moving in the air for a tethered fly.n order to validate the realism of the generated simulated walking trajectories, the authors compare various attributes of simulated to real tethered fly trajectories and show qualitative and quantitative similarities, including using a novel metric coined as Kinematic Similarity (KS). The KS score of a trajectory is a measure of the likelihood that the trajectory belongs to the distribution of real trajectories estimated from the experimental data. While such a metric is a useful tool to validate the quality of simulated data, there is some room for improvement in the actual computation of this score. For instance, the KS score is computed for any given time-window of walking simulation using a fraction of information from the joint-angle trajectories. It is unclear if the remaining information in joint-angle trajectories that are not used in the computation of the KS score can be ignored in the context of validating the realism of simulated walking trajectories.The authors validate simulated walking trajectories generated by the trained model under a range of sensorimotor delays and external perturbations. The trained model is shown to generate realistic jointangle trajectories in the presence of external perturbations as long as the sensorimotor delays are constrained within a certain range. This range of sensorimotor delays is shown to be comparable to experimental measurements of sensorimotor delays, leading to the conclusion that the fly nervous system is just fast enough to be robust to perturbations.Strengths:This work presents a novel framework to simulate *Drosophila* walking in the presence of external perturbations and sensorimotor delay. Although the model makes some simplifying assumptions, it has sufficient complexity to generate new, testable hypotheses regarding motor control in *Drosophila*. The authors provide evidence for realistic simulated walking trajectories by comparing simulated trajectories generated by their trained model with experimental data using a novel metric proposed by the authors. The model proposes a crucial role in future predictions to ensure robust walking trajectories against external perturbations and motor delay. Realistic simulations under a range of prediction intervals, perturbations, and motor delays generating realistic walking trajectories support this claim. The modular architecture of the framework provides opportunities to make testable predictions regarding motor control in *Drosophila*. The work can be of interest to the *Drosophila* community interested in digitally simulating realistic models of *Drosophila* locomotion behaviors, as well as to experimentalists in generating testable hypotheses for novel discoveries regarding neural control of locomotion in *Drosophila*. Moreover, the work can be of broad interest to neuroethologists, serving as a benchmark in modelling animal locomotion in general.

We thank the reviewer for their positive comments.

Weaknesses:As the authors acknowledge in their work, the control and dynamics model makes some simplifying assumptions about *Drosophila* physics/physiology in the context of walking. For instance, the model does not incorporate ground contact forces and inertial effects of the fly's body. It is not clear how these simplifying assumptions would affect some of the quantitative results derived by the authors. The range of tolerable values of sensorimotor delays that generate realistic walking trajectories is shown to be comparable with sensorimotor delays inferred from physiological measurements. It is unclear if this comparison is meaningful in the context of the model's simplifying assumptions.

We now discuss how some of these assumptions affect the quantitative results in the section “Towards biomechanical and neural realism”. We reproduce the relevant sentences below:

“The inclusion of explicit leg-ground contact interactions would also make it harder for the model to recover when perturbed, because perturbations during walking often occur upon contact with the ground (e.g. the ground is slippery or bumpy).”

“We anticipate that the increased sensory resolution from more detailed proprioceptor models and the stability from mechanical compliance of limbs in a more detailed biomechanical model would make the system easier to control and increase the allowable range of delay parameters. Conversely, we expect that modeling the nonlinearity and noise inherent to biological sensors and actuators may decrease the allowable range of delay parameters.”

The authors propose a novel metric coined as Kinematic Similarity (KS) to distinguish realistic walking trajectories from unrealistic walking trajectories. Defining such an objective metric to evaluate the model's predictions is a useful exercise, and could potentially be applied to benchmark other computational animal models that are proposed in the future. However, the KS score proposed in this work is calculated using only the first two PCA modes that cumulatively account for less than 50% of the variance in the joint angles. It is not obvious that the information in the remaining PCA modes may not change the log-likelihood that occurs in the real walking data.

The primary reason we designed the KS metric was to determine whether the simulated fly continues walking in the presence of perturbations. We initially limited the analysis of the KS to the first 2 principal components. For completeness, we now investigate the additional principal components in Appendix 8 and the effect of evaluating KS with different numbers of components in Appendix 9.

Overall, the results look similar when including additional components for impulse perturbations. For stochastic perturbations, the range of similar walking decreases as we increase the number of components used to evaluate walking kinematics. Comparing this with Appendix 8, which shows that higher components represent higher frequencies of the walking cycle, we conclude that at the edge of stability for delays (where sum of sensory and actuation delays are about 40ms), flies can continue walking but with impaired higher frequencies (relative to no perturbations) during and after perturbation.

We added the following text in the methods:

“We chose 2 dimensions for PCA for two key reasons. First, these 2 dimensions alone accounted for a large portion of the variance in the data (52.7% total, with 42.1% for first component and 10.6% for second component). There was a big drop in variance explained from the first to the second component, but no sudden drop in the next 10 components (see Appendix 8). Second, the KDE procedure only works effectively in low-dimensional spaces, and the minimal number of dimensions needed to obtain circular dynamics for walking is 2. We investigate the effect of varying the number of dimensions of PCA in Appendix 9.”

(Note that we have corrected the percentage of variance accounted for by the principal components, as these numbers were from an older analysis prior to the first draft.)

We also reference Appendix 9 in the results:

“We observed that robust walking was not contingent on the specific values of motor and sensory delay, but rather the sum of these two values (Fig. 5E). Furthermore, as delay increases, higher frequencies of walking are impacted first before walking collapses entirely (Appendix 9).”

**Reviewer #2 (Public Review):**
Summary:In this study, Karashchuk et al. develop a hierarchical control system to control the legs of a dynamic model of the fly. They intend to demonstrate that temporal delays in sensorimotor processing can destabilize walking and that the fly's nervous system may be operating with as long of delays as could possibly be corrected for.Strengths:Overall, the approach the authors take is impressive. Their model is trained using a huge dataset of animal data, which is a strength. Their model was not trained to reproduce animal responses to perturbations, but it successfully rejects small perturbations and continues to operate stably. Their results are consistent with the literature, that sensorimotor delays destabilize movements.Weaknesses:The model is sophisticated and interesting, but the reviewer has great concerns regarding this manuscript's contributions, as laid out in the abstract:(1) Much simpler models can be used to show that delays in sensorimotor systems destabilize behavior (e.g., Bingham, Choi, and Ting 2011; Ashtiani, Sarvestani, and Badri-Sproewitz 2021), so why create this extremely complex system to test this idea? The complexity of the system obscures the results and leaves the reviewer wondering if the instability is due to the many, many moving parts within the model. The reviewer understands (and appreciates) that the authors tested the impact of the delay in a controlled way, which supports their conclusion. However, the reviewer thinks the authors did not use the most parsimonious model possible, and as such, leave many possible sources for other causes of instability.

We thank the reviewer for this observation — we agree that we did not make the goal of the work quite clear. The goal of this paper was to build an interpretable and generalizable model of fly walking, which was then used to investigate varying sensorimotor delays in the context of locomotion. To this end, we used a modular model to recreate walking kinematics, and then investigated the effect of delays on locomotion. Locomotion in itself is a complex phenomenon — thus, we have chosen a model that is complex enough to reasonably recapitulate joint trajectories, while remaining interpretable.

We have clarified this in the text near the end of the introduction:

“Here, we develop a new, interpretable, and generalizable model of fly walking, which we use to investigate the impact of varying sensorimotor delays in *Drosophila* locomotion.”

We also emphasize the investigation of sensorimotor delays in the context of locomotion in the beginning of the “Effect of sensory and motor delays on walking” section:

“... we used our model to investigate how changing sensory and motor delays affects locomotor robustness.”

We also remark that while they are very relevant papers for our work, neither of the prior papers focus on locomotion: the first involves a 2D balance model of a biped, and the second involves drop landings of quadrupeds.

Lastly, we note that the investigation of delay is not the only use for this model — in the future, this model can also be used to study other aspects of locomotion such as the role of proprioceptive feedback (see “Role of proprioceptive feedback in fly walking” section). The layered framework of the model can also be extended to other animals and locomotor strategies (see “Layered model produces robust walking and facilitates local control” section”).

(2) In a related way, the reviewer is not sure that the elements the authors introduced reflect the structure or function of the fly's nervous system. For example, optimal control is an active field of research and is behind the success of many-legged robots, but the reviewer is not sure what evidence exists that suggests the fly ventral nerve cord functions as an optimal controller. If this were bolstered with additional references, the reviewer would be less concerned.

We thank the reviewer for the comment — we have now further clarified how our model elements reflect the fly’s nervous system. The elements we introduce are plausible but only loosely analogous to the fly’s nervous system. While we draw parallels from these elements to anatomy (e.g. in Fig 1A-B, and in the first paragraph of the Results section), we do not mean to suggest that these functional elements directly correspond to specific structures in the fly’s nervous system. A substantial portion of the suggested future work (see “Towards biomechanical and neural realism”) aims to bridge the gap between these functional elements and fly physiology, which is beyond the scope of this work.

We have added clarifying text to the Results section:

“While the model is inspired by neuroanatomy, its components do not strictly correspond to components of the nervous system --- the construction of a neuroanatomically accurate model is deferred to future work (see Discussion).”

In the specific case of optimal control — optimal control is a theoretical model that predicts various aspects of motor control in humans, there is evidence that optimal control is implemented by the human nervous system (Todorov and Jordan, 2002; Scott, 2004; Berret et al., 2011). Based on this, we make the assumption that optimal control is a reasonable model for motor control in flies implemented by the fly nervous system as well. Fly movement makes use of proprioceptive feedback signals (Mendes et al., 2013; Pratt et al., 2024; Berendes et al., 2016), and optimal control is a plausible mechanism that incorporates feedback signals into movement.

We have added the following clarifying text in the Results section:

“The optimal controller layer maintains walking kinematics in the presence of sensori motor delays and helps compensate for external perturbations. This design was inspired by optimal control-based models of movements in humans (Todorov and Jordan, 2002; Scott, 2004; Berret et al., 2011)”

(3) "The model generates realistic simulated walking that matches real fly walking kinematics...". The reviewer appreciates the difficulty in conducting this type of work, but the reviewer cannot conclude that the kinematics "match real fly walking kinematics". The range of motion of several joints is 30% too small compared to the animal (Figure 2B) and the reviewer finds the video comparisons unpersuasive. The reviewer would understand if there were additional constraints, e.g., the authors had designed a robot that physically could not complete the prescribed motions. However the reviewer cannot think of a reason why this simulation could not replicate the animal kinematics with arbitrary precision, if that is the goal.

We agree with the reviewer that the model-generated kinematics are not perfectly indistinguishable from real walking kinematics, and now clarify this in the text. We also agree with the reviewer that one could build a model that precisely replicates real kinematics, but as they intuit, that was not our goal. Our goal was to build a model that both replicates animal kinematics, and is interpretable and generalizable (which allows us to investigate what happens when perturbations and varying sensorimotor delays are introduced). There is a trade-off between realism and generalizability — a simulation that fully recreates empirical data would require a model that is completely fit to data, which is likely to be more complex (in terms of parameters required) and less generalizable to novel scenarios. We have made design choices that result in a model that balances these trade-offs. We do not consider this to be a weakness of the model; in fact, few comparable models account for all joints involved in locomotion, and fewer explicitly compare model kinematics with kinematics from data.

We have tempered the language in the abstract:

“The model generates realistic simulated walking that resembles real fly walking kinematics”

The tempered statement, we believe, is a fair characterization of the walking — it resembles but does not perfectly match real kinematics.

We have also introduced clarifying text in the introduction:

“Overall, existing walking models focus on either kinematic or physiological accuracy, but few achieve both, and none consider the effect of varying sensorimotor delays. Here, we develop a new, interpretable, and generalizable model of fly walking, which we use to investigate the impact of varying sensorimotor delays in *Drosophila* locomotion.”

**Recommendations for the authors:**

**Reviewer #1 (Recommendations For The Authors):**
Potential typo on page 5:2.1.2 Joint kinematics trajectory generatorParagraph 4, last line: Original text - ".....it also estimates the current phase". Suggested correction - "...it also estimates the current phase velocity"

Done

Potential typo on page 8:2.3 Model maintains walking under unpredictable external perturbations.Paragraph 3, line 2: Original text - "...brief, unexpected force (e.g. legs slipping on an unstable surface)".Consider replacing force with motion, or providing an example of a force as opposed to displacement (slipping).

Done

Potential typo on page 8:2.3 Model maintains walking under unpredictable external perturbations.Paragraph 3, line 4: Original text - "The magnitude of this velocity is drawn from a normal distribution...".Is this really magnitude? If so, please discuss how the sign (+/-) is assigned to velocity, and how the normal distribution is centred so as to sample only positive values representing magnitude.

Indeed the magnitude of the velocity is drawn from a normal distribution. A positive or negative sign is then assigned with equal odds. We have added text to clarify this:

“The sign of the velocity was drawn separately so that there is equal likelihood for negative or positive perturbation velocities.”

Page 8:2.3 Model maintains walking under unpredictable external perturbations.In Paragraph 5: Why is the data reduced to only 2 dimensions? Could higher order PCA modes (cumulatively accounting for more than 50% variance in the data) not have distinguishing information between realistic and unrealistic walking trajectories?

We provide a longer response for this in the public review above.

Page 11:Why wouldn't a system trained in the presence of external perturbations perform better? What is the motivation to remove external perturbations during training?

We agree that a system trained in the presence of external perturbations would probably perform better — however, we do not have data that contains walking with external perturbations. Nothing was removed — all the data used in this study involve a fly walking without perturbations.

We have added a clarification:

“our model maintains realistic walking in the presence of external dynamic perturbations, despite being trained only on data of walking without perturbations (no perturbation data was available).”

Page 16:4.1 Tracking joint angles of *D. melanogaster* walking in 3D.Paragraph 1: Readers who wish to collect similar data might benefit from specifying the exposure time, animal size in pixels (or camera sensor format and field of view), in addition to the frame rate. Alternatively, consider mentioning the camera and lens part numbers provided by the manufacturer.

This is a good point. We have updated the text to include these specifications:

“We obtained fruit fly *D. melanogaster* walking kinematics data following the procedure previously described in (Karashchuk et al, 2021). Briefly, a fly was tethered to a tungsten wire and positioned on a frictionless spherical treadmill ball suspended on compressed air. Six cameras (Basler acA800-510um with Computar zoom lens MLM3X-MP) captured the movement of all of the fly's legs at 300 Hz. The fly size in pixels ranges from about 300x300 up to 700x500 pixels across the 6 cameras. Using Anipose, we tracked 30 keypoints on the fly, which are the following 5 points on each of the 6 legs: body-coxa, coxa-femur, femur-tibia, and tibia-tarsus joints, as well as the tip of the tarsus.”

Potential typos on page 18:4.3.3 Training procedureParagraph 2, line 1: Original text - "..(, p)"Do the authors mean "...(,)"Paragraph 2, line 2: Original text - "... (,, v, p)" Do the authors mean "... (,, v,)"?Paragraph 3, line 3: Original text - "... (,, v, p)" Do the authors mean "... (,, v,)"?

Thank you for pointing out this issue. We have now fixed the phase p to be \phi to be consistent with the rest of the text.

Paragraph 3, line 3: Original text - "...()"Do the authors mean "(d)"? If not, please discuss the difference between and d.

Thank you for pointing this out. \hat \theta and \theta_d were used interchangeably which is confusing. We have standardized our reference to the desired trajectory as \theta_d throughout the text.

Page 19:Typo after eqn. (6):Original text: "where x : = q - q, ... A and B are Jacobians with respect to...."Correction: "where x : = q - q, ... Ac and Bc are Jacobians with respect to...."Similar corrections in eqn. 7 and eqn. 8: A and B should be replaced with Ac and Bc. Done

Page 19, eqn. (10b):

Should the last term be qd(t+T) as opposed to qd(t+1)?

No: in fact (10a) contains the typo: it should be y(t+1) as opposed to y(t+T). This has been fixed.

Page 19The authors' detailed description of the initial steps leading up to the dynamics model, involving the construction of the ODE, linearizing the system about the fixed point makes the text broadly accessible to the general reader. Similarly, adding some more description of the predictive model (eqn. 11 - 15) could improve the text's accessibility and the reader's appreciation for the model. This is especially relevant since the effects of sensorimotor delay and external perturbations, which are incorporated in the control and dynamics model, form a major contribution to this work. What do the matrices F, G, L, H, and K look like for the *Drosophila* model? Are there any differences between the model in Stenberg et al. (referenced in the paper) and the authors' model for predictive control? Are there any differences in the assumptions made in Stenberg et al. compared to the model presented in this work? The readers would likely also benefit from a figure showing the information flow in the model, and describing all the variables used in the predictive control model in eqn. 11 through eqn. 15 (analogous to Figure 1 in Stenberg et al. (2022)). Such a detailed description of the control and dynamics model would help the reader easily appreciate the assumptions made in modelling the effects of sensorimotor delay and external perturbations.

Done

Page 20:Eqn. 12: Should z(t+1) be z(t+T) instead?Similar comment for eqn. 14

No: we made a mistake in (10a); there should be no (t+T) terms; all terms should be (t+1) terms to reflect a standard discrete-time difference equation.

Eqn. 13: r(t) can be defined explicitly

Done

4.5 Generate joint trajectories of the complete model with perturbations Paragraph 2, line 2: Please read the previous comment

\hat \theta and \theta_d were previously used interchangeably which is confusing. We have standardized our reference to the desired trajectory as \theta_d throughout the text.

Original text - "Every 8 timesteps, we set :=...."Does this mean dis set to? If so, the motivation for this is not clear.

We mean that \theta_d is set to be equal to \theta. We have replaced “:=” with “=” for clarity.

General comments for the authors:Could the authors discuss the assumptions regarding *Drosophila* physiology implied in the control model?

The control model is primarily included as a plausible functional element of the fly’s nervous system, and as such implies minimal assumptions on physiology itself. The main assumption, which is evident from the description of the model components, is that the fly uses proprioceptive feedback information to inform future movements.

We have added clarifying text to the Results section:

“While the model is inspired by neuroanatomy, its components do not strictly correspond to components of the nervous system --- the construction of a neuroanatomically accurate model is deferred to future work (see Discussion).”

The authors acknowledge the absence of ground contact forces in the model. It is probably worth discussing how this simplification may affect inferences regarding the acceptable range of sensorimotor delay in generating realistic walking trajectories.

We agree, and discuss how some of these assumptions affect the quantitative results in the section “Towards biomechanical and neural realism”. We replicate the relevant sentences below:

“The inclusion of explicit leg-ground contact interactions would also make it harder for the model to recover when perturbed, because perturbations during walking often occur upon contact with the ground (e.g. the ground is slippery or bumpy).”

The effects of other simplifications are also mentioned in the same section.

Can the authors provide an insight into why the use of a second derivative of joint angles as the output of the trajectory generator () leads to more realistic trajectories (4.3.1 Model formulation, paragraph 1)?Does the use of a second-order derivative of joint angles lead to drift error because of integration?Could the distribution of θd produced be out of the domain due to drift errors? Could this affect the performance of the neural network model approximating the trajectory generator?

We are not sure why the second derivative works better than the first derivative. It is possible that modeling the system as a second order differential equation gives the network more ability to produce complex dynamics.

As can be seen in the example time series in Figures 2 and 3 and supplemental videos, there is no drift error from integration, so it is unlikely to affect the performance of the neural network.

What does the model's failure (quantified by a low KS score) look like in the context of fly dynamics? What do the joint angles look like for low values of KS score? Does the fly fall down, for example?

Since the model primarily considers kinematics, a low KS score means that kinematics are unrealistic, e.g. the legs attain unnatural angles or configurations. Examples of this can be seen in videos 4-7, as well as in the bottom row of Fig. 5, panel A. Here, at 40ms of motor delay, L2 femur rotation is seen to attain values that far exceed the normal ranges.

We have added a small clarification in the caption of Fig.5 panel A:

“low KS indicates that the perturbed walking deviates from data and results in unnatural angles

(as seen at 40ms motor delay)”

We remark that since our simulations do not incorporate contact forces (as the reviewer remarks above, we simulate something like legs moving in the air for a tethered fly), the fly cannot “fall down” per se. However, if forces were incorporated then yes, these unrealistic kinematics would correspond to a fly that falls down or is no longer walking.

**Reviewer #2 (Recommendations For The Authors):**
L49: "Computational models of locomotion do not typically include delay as a tunable parameter, and most existing models of walking cannot sustain locomotion in the presence of delays and external perturbations". This remark confuses the reviewer.(1) If models do not "typically" include delay as a tunable parameter, this suggests that atypical models do. Which models do? Please provide references.

Our initial phrasing was confusing. We meant to say that most models do not include delay, and some models do include delay as a fixed value (rather than a tunable value). We clarify in the updated text, which is replicated below:

“Computational models of locomotion typically have not included delays as a tunable parameter, although some models have included them as fixed values (Geyer and Herr, 2010; Geijtenbeek et al., 2013).”

(2) Has the statement that most existing models cannot sustain locomotion with delays been tested? If so, provide references. If not, please remove this statement or temper the language.

Since most models don’t include delays, they cannot be run in scenarios with delays. We clarify in the updated text, which is replicated below:

“Computational models of locomotion have not typically included delays. Some have included delay as a fixed value rather than a tunable parameter (Geyer and Herr, 2010; Geijtenbeek et al., 2013). However, in general, the impact of sensorimotor delays on locomotor control and robustness remains an underexplored topic in computational neuroscience.”

L57: "two of six legs lift off the ground at a time" - Two legs are off the ground at any time, but they do not "lift off" simultaneously in the fruit fly. To lift off simultaneously, contralateral leg pairs would need to be 33% out of phase with one another, but they are almost always 50% out of phase.

Thank you for pointing out this oversight. We have updated the text accordingly:

“Flies walk rhythmically with a continuum of stepping patterns that range from tetrapod (where two of six legs are off the ground at a time) to tripod (where three of six legs are off the ground at a time)"

L88: "a new model of fly walking" - The intention of the authors is to produce a model from which to learn about walking in the fly, is that correct? The reviewer has read the paper several times now and wants to be sure that this is the authors' goal, not to engineer a control system for an animation or a robot.

Indeed, this is our goal. We were previously unclear about this, and have made text edits to clarify this — we provide a longer response for this in the public review above (see (1)).

L126: "These desired phases are synchronized across pairs of legs to maintain a tripod coordination pattern, even when subject to unpredictable perturbations." - Does the animal maintain tripod coordination even when perturbed? In the reviewer's experience, flies vary their interleg coordination all the time. The reviewer would also expect that if perturbed strongly (as the supplemental videos show), the animal would adapt its interleg coordination in response. The author finds this assumption to be a weak point in the paper for the use of this disturbance exploring animal locomotion.

We do not know exactly how flies may react to our mechanical perturbations. However, we may hypothesize based on past papers.

Couzin-Fuchs et al (2015) apply a mechanical perturbation to walking cockroaches. They find that that tripod is temporarily broken immediately after the perturbation but the cockroach recovers to a full tripod within one step cycle.

DeAngelis et al (2019) apply optogenetic perturbations to fly moonwalker neurons that drive backward walking. Flies slow down following perturbation, but then recover after 200ms (about 2-3 steps) to their original speed (on average).

Thus, we think it is reasonable to model a fly’s internal phase coupling to maintain tripod and for its intended speed to remain the same even after a perturbation.

We do agree with the reviewer that it is plausible a fly might also slow down or even stop after a perturbation and we do not model such cases. We have added some text to the discussion on future work:

“Future work may also model how higher-level planning of fly behavior interacts with the lowerlevel coordination of joint angles and legs. Walking flies continuously change their direction and speed as they navigate the environment (Katsov et al, 2017; Iwasaki et al 2024). Past work shows that flies tend to recover and walk at similar speeds following perturbations (DeAngelis et al, 2019), but individual flies might still change walking speed, phase coupling, or even transition to other behaviors, such as grooming. Modeling these higher-level changes in behavior would involve combining our sensorimotor model with models for navigation (Fisher 2022) or behavioral transitions (Berman et al, 2016).”

L136: "...to output joint torques to the physical model of each leg" - Is this the ultimate output of the nervous system? Muscles are certainly not idealized torque generators. There are dynamics related to activation and mechanics. The reviewer is skeptical that this is a model of neural control in the animal, because the computation of the nervous system would be tuned to account for all these additional dynamics.

We agree with the reviewer that joint torques are not the ultimate output of the nervous system. We use a torque controller because it is parsimonious, and serves our purpose of creating an interpretable and modular locomotion model.

We also agree that muscles are an important consideration — we make mention of them later on in the paper under the section “Toward biomechanical and neural realism”, where we state “Another step toward biological realism is the incorporation of explicit dynamical models of proprioceptors, muscles, tendons, and other biomechanical aspects of the exoskeleton.”

Our goal is not to directly model neural control of the animal. We have introduced text clarifications to emphasize this — we provide a longer response for this in the public review above (see (2)).

L143: "To train the network from data, we used joint kinematics of flies walking on a spherical treadmill..." This is an impressive approach, but then the reviewer is confused about why the kinematics of the model are so different from those of the animal. The animal takes longer strides at a lower frequency than the model. If the model were trained with data, why aren't they identical? This kind of mismatch makes the reviewer think the approach in this paper is too complicated to address the main problem.

The design of our trajectory generator model is one of the simplest for reproducing the output of a dynamical system. It consists of a multilayer perceptron model that models the phase velocity and joint angle accelerations at each timestep. All of its inputs are observable and interpretable: the current joint angles, joint angle derivatives, desired walking speed, and phase angle.

We chose this model for ease of interpretability, integration with the optimal controller, and to allow for generalization across perturbations. Given all of these constraints, this is the best model of desired kinematics we could obtain. We note that the simulated kinematics do match real fly kinematics qualitatively (Figure 2A and supplemental videos) and are close quantitatively (Figure 2B and C). We speculate that matching the animals’ strides at all walking frequencies may require explicitly modeling differences across individual flies. We leave the design and training of more accurate (but more complex) walking models for future work.

We add some further discussion about fitting kinematics in the discussion:

“Although we believe our model matches the fly walking sufficiently for this investigation, we do note that our model still underfits the joint angle oscillations in the walking cycle of the fly (see Figure 2 and Appendix 2). More precise fitting of the joint angle kinematics may come from increasing the complexity of the neural network architecture, improving the training procedure based on advances in imitation learning (Hussein et al., 2018), or explicitly accounting for individual differences in kinematics across flies (Deangelis et al., 2019; Pratt et al., 2024).”

Figure 2: The reviewer thinks the violin plots in Figure 2C are misleading. Joint angles could be greater or less than 0, correct? If so, why not keep the sign (pos/neg) in the data? Taking the absolute value of the errors and "folding over" the distribution results in some strange statistics. Furthermore, the absolute value would shroud any systematic bias in the model, e.g., joint angles are always too small. The reviewer suggests the authors plot the un-rectified data and simply include 2 dashed lines, one at 5.56 degrees and one at -5.56 degrees.

These violin plots are averages of errors over all phases within each speed. We chose to do this to summarize the errors across all phase angle plots, which are shown in detail in Appendix 2 and 3.

For the reviewer, we have added a plot of the raw errors across all phase angle plots in Appendix 4, E.

L156: Should "\phi\dot" be "\phi"?

We originally had a typo: we said “phase” when we meant “phase velocity”. This has been fixed. \phi\dot is correct.

L160: "This control is possible because the controller operates at a higher temporal frequency than the trajectory generator...". This statement concerns the reviewer. To the reviewer, this sounds like the higher-level control system communicates with the "muscles" at a higher frequency than the low-level control system, which conflicts with the hierarchical timescales at which the nervous system operates. Or do the authors mean that the optimal controller can perform many iterations in between updates from the trajectory generator level? If so, please clarify.

We mean that the optimal controller can perform many iterations in between updates from the trajectory generator level. The text has been clarified:

“This control is possible because the controller operates at a higher temporal frequency than the trajectory generator in the model. The controller can perform many iterations (and reject disturbances) in between updates to and from the trajectory generator.”

L225: "We considered two types of perturbations: impulse and persistent stochastic". Are these realistic perturbations? Realistic perturbations such as a single leg slipping, or the body movement being altered would produce highly correlated joint velocities.

These perturbations are not quite realistic — nonetheless, we illustrate their analogousness to real perturbations in the subsequent text in the paper, and restrict our simulations to ranges that would be biologically plausible (see Appendix 6). We agree that realistic perturbations would produce highly correlated joint accelerations and velocities, whereas our perturbations produce random joint accelerations.

L265: "...but they are difficult to manipulate experimentally..." This is true, but it can and has been done. The authors should cite:

Bässler, U. (1993). The femur-tibia control system of stick insects-A model system for the study of the neural basis of joint control. Brain Research Reviews, 18(2), 207-226.

Thank you for the suggestion, we have incorporated it into the text at the end of the referenced sentence.

L274: "...since the controller can effectively compensate for large delays by using predictions of joint angles in the future". But can the nervous system do this? Or, is there a reason to think that the nervous system can? The reviewer thinks the authors need stronger justification from the literature for their optimal control layer.

To clarify, this sentence describes a feature of the model’s behavior when no external perturbations are present. This is not directly relevant to the nervous system, since organisms do not typically exist in an environment free of perturbations — we are not suggesting that the nervous system does this.

In response to the question of whether the nervous system can compensate for delays using predictions: we know that delays are present in the nervous system, perturbations exist in the environment, and that flies manage to walk in spite of them. Thus, some type of compensation must exist to offset the effects of delays (the reviewer themself has provided some excellent citations that study the effects of delays). In our model, we use prediction as the compensation mechanism — this is one of our central hypotheses. We further discuss this in the section “Predictive control is critical for responding to perturbations due to motor delay”.

L319: "The formulation of a modular, multi-layered model for locomotor control makes new experimentally-testable hypotheses about fly motor control...". What testable hypotheses are these? The authors should explicitly state them. They are not clear to the reviewer, especially given the nonphysiological nature of the control system and the mechanics.

A number of testable hypotheses are mentioned throughout the Discussion section:

“Our model predicts that at the same perturbation magnitude, walking robustness decreases as delays increase. This could be experimentally tested by altering conduction velocities in the fly, for example by increasing or decreasing the ambient temperature (Banerjee et al, 2021). If a warmer ambient temperature decreases delays in the fly, but fly walking robustness remains the same in response to a fixed perturbation, this would indicate a stronger role for central control in walking than our modeling results suggest.”

“In our model, robust locomotion was constrained by the cumulative sensorimotor delay. This result could be experimentally validated by comparing how animals with different ratios of sensory to motor delays respond to perturbations. Alternatively, it may be possible to manipulate sensory vs. motor delays in a single animal, perhaps by altering the development of specific neurons or ensheathing glia (Kottmeier et al., 2020). If sensory and motor delays have significantly different effects on walking quality, then additional compensatory mechanisms for delays could play a larger role than we expect, such as prediction through sensory integration, mechanical feedback, or compensation through central control.”

“we hypothesize that removing proprioceptive feedback would impair an insect's ability to sustain locomotion following external perturbations.”

“We propose that fly motor circuits may encode predictions of future joint positions, so the fly may generate motor commands that account for motor neuron and muscle delays.”

L323: "...and biomechanical interactions between the limb and the environment". In the reviewer's experience, the primary determinant of delay tolerance is the mechanical parameters of the limb: inertia, damping, and parallel elasticity. For example, in Ashtiani et al. 2021, equation 5 shows exactly how this comes about: the delay changes the roots and poles of the control system. This is why the reviewer is confused by the complexity of the model in this submission; a simpler model would explain why delays cannot be tolerated in certain circumstances.

We were previously unclear about the goal of the model, and have made text edits to clarify this — we provide a longer response for this in the public review above (see (1)).

L362: Another highly relevant reference here would be Sutton et al. 2023.

Done

L366: Szczecinski et al. 2018 is hardly a "model"; it is mostly a description of experimental data. How about Goldsmith, Szczecinski, and Quinn 2020 in B&B? Their model of fly walking has patterngenerating elements that are coordinated through sensory feedback. In their model, motor activation is also altered by sensory feedback. The reviewer thinks the statement "Models of fly walking have ignored the role of feedback" is inaccurate and their description of these references should be refined.

Thank you for the suggestion; we have tempered the language and revised this section to include more references, including the suggested one — text is replicated below.

“Many models of fly walking ignore the role of feedback, relying instead on central pattern generators (Lobato-Rios et al., 2022; Szczecinski et al., 2018; Aminzare et al., 2018) or metachondral waves (Deangelis et al., 2019) to model kinematics. Some models incorporate proprioceptive feedback, primarily as a mechanism that alters timing of movements in inter-leg coordination (Goldsmith et al., 2020; Wang-Chen et al., 2023).”

We remark that Szczecinski et al does include a model that replicates data without using sensory feedback, so we think it is fair to include.

L371: "...highly dependent on proprioceptive feedback for leg coordination during walking." What about Berendes et al. 2016, which showed that eliminating CS feedback from one leg greatly diminished its ability to coordinate with the other legs? This suggests that even flies depend on sensory feedback for proper coordination, at least in some sense.

Interesting suggestion – we have integrated it into the text a little further down, where it better fits:

“Silencing mechanosensory chordotonal neurons alters step kinematics in walking *Drosophila* (Mendes et al., 2013; Pratt et al., 2024). Additionally, removing proprioceptive signals via amputation interferes with inter-leg coordination in flies at low walking speeds (Berendes et al., 2016)”

L426: "The layered model approach also has potential applications for bio-mimetic robotic locomotion.". How fast can this model be computed? Can it run faster than real-time? This would be an important prerequisite for use as a robot control system.

The model should be able to be run quite fast, as it involves only

(1) Addition, subtraction, matrix multiplication, and sinusoidal computation on scalars (for the phase coordinator and optimal controller)

(2) Neural network inference with a relatively small network (for the trajectory generator) Whether this can run in real-time depends on the hardware capabilities of the specific robot and the frequency requirements — it is possible to run this on a desktop or smaller embedded device.

We do note that the model needs to first be set up and trained before it can be run, which takes some time (see panel D of Figure 1).

L432: "...which is a popular technique in robotics.". Please cite references supporting this statement.

We have added citations: the text and relevant citations are reproduced below:

“... which is a popular technique in robotics (Hua et al., 2021; Johns, 2021)

Hua J, Zeng L, Li G, Ju Z. Learning for a robot: Deep reinforcement learning, imitation learning, transfer learning. Sensors. 2021; 21(4):1278

Johns E. Coarse-to-fine imitation learning: Robot manipulation from a single demonstration. In:

2021 IEEE international conference on robotics and automation (ICRA) IEEE; 2021. p. 4613–4619

L509: "We find that the phase offset across legs is not modulated across walking speeds in our dataset". This is a surprising result to the reviewer. Looking at Figure 6C, the reviewer understands that there are no drastic changes in coordinate with speed, but there are certainly some changes, e.g., L1-R3, L3-R1. In the reviewer's experience, even very small changes in interleg phasing can change the visual classification of walking from "tripod" to "tetrapod" or "metachronal". Furthermore, several leg pairs do not reside exactly at 0 or \pi radians apart, e.g., L1-L3, L2-L3, R1-R3, R2-R3. In conclusion, the reviewer thinks that setting the interleg coordination to tripod in all cases is a large assumption that requires stronger justification (or, should be eliminated altogether).

We made a simplifying assumption of a tripod coordination across all speeds. The change in relative phase coordination across speeds is indeed relatively small and additionally we see little change in our results across forward speeds (see Figures 4B, 5C and 5D).

We have added text to clarify this assumption and what could be changed for future studies in the methods:

“We estimate $\bar \phi_{ij}$ from the walking data by taking the circular mean over phase differences of pairs the legs during walking bouts. We find that the phase offset across legs is not strongly modulated across walking speeds in our dataset (see Appendix 1) so we model $\bar \phi_{ij}$ as a single constant independent of speed. In future studies, this could be a function of forward and rotation speeds to account for fine phase modulation differences.”

L581: "of dimension...". Should the asterisk be replaced by \times? The asterisk makes the reviewer think of convolution. This change should be made throughout this paragraph.

Good point, done.

Figure 6: Rotational velocities in all 3 sections are reported in mm/s, but these units do not make sense. Rotational velocities must be reported in rad/s or deg/s.

The rotation velocity of mm/s corresponded to the tangential velocity of the ball the fly walked on. We agree that this does not easily generalize across setups, so we have updated the figure rotation velocities in rad/s.

L619: The reviewer is unconvinced by using only 2 principal components of the data to compare the model and animal kinematics. The authors state on line 626 that the 2 principal components do not capture 56.9% of the variation in the data, which seems like a lot to the reviewer. This is even more extreme considering that the model has 20 joints, and the authors are reducing this to 2 variables; the reviewer can't see how any of the original waveforms, aside from the most fundamental frequencies, could possibly be represented in the PCA dataset. If the walking fly models looked similar to each other, the reviewer could accept that this method works. But the fact that this method says the kinematics are similar, but the motion is clearly different, leads the reviewer to suspect this method was used so the authors could state that the data was a good match.

Our primary use of the KS metric was to indicate whether the simulated fly continues walking in the presence of perturbations, hence we limited the analysis of the KS to the first 2 principal components.

For completeness, we investigate the principal components in Appendix 8 and the effect of evaluating KS with different numbers of components in Appendix 9.

The results look similar across components for impulse perturbations. For stochastic perturbations, the range of similar walking decreases as we increase the number of components used to evaluate walking kinematics. Comparing this with Appendix 8 showing that higher components represent higher frequencies of the walking cycle, we conclude that at the edge of stability for delays (where sum of sensory and actuation delays are about 40ms), flies can continue walking but with impaired higher frequencies (relative to no perturbations) during and after perturbation.

We add text in the methods:

“We chose 2 dimensions for PCA for two key reasons. First, these 2 dimensions alone accounted for a large portion of the variance in the data (52.7% total, with 42.1% for first component and 10.6% for second component). There was a big drop in variance explained from the first to the second component, but no sudden drop in the next 10 components (see Appendix 8). Second, the KDE procedure only works effectively in low-dimensional spaces, and the minimal number of dimensions needed to obtain circular dynamics for walking is 2. We investigate the effect of varying the number of dimensions of PCA in Appendix 9.”

(Note that we have corrected the percentage of variance accounted for by the principal components, as these numbers were from an older analysis prior to the first draft.)

We also reference Appendix 9 in the results:

“We observed that robust walking was not contingent on the specific values of motor and sensory delay, but rather the sum of these two values (Fig. 5E). Furthermore, as delay increases, higher frequencies of walking are impacted first before walking collapses entirely (Appendix 9).”